

**Cross-verification of simulated GEMS tropospheric**
**ozone retrievals and ozonesonde measurements**
**Over Northeast Asia**
**Juseon Bak**[1,#] (Juseon.bak@cfa.harvard.edu)
**Kang-Hyeon Baek**[1] (iambk100@gmail.com) **Jae-Hwan Kim**[1, *] (jakim@pusan.ac.kr)
**Xiong Liu**[2] (xliu@cfa.harvard.edu)
**Jhoon Kim**[3] (jkim2@yonsei.ac.kr) **Kelly Chance**[2] (kchance@cfa.harvard.edu)
[1)] Atmospheric Science Department, Pusan National University, Busan, Korea
[2)] Atomic and Molecular Physics Division, Harvard-Smithsonian Center for Astrophysics, Cambridge, USA.
[3)] Department of Atmospheric Sciences, Yonsei University, Seoul, Korea
[#]Currently at Atomic and Molecular Physics Division, Harvard-Smithsonian Center for Astrophysics, Cambridge, USA.
[*]Corresponding Author



**Abstract**

The Geostationary Environment Monitoring Spectrometer (GEMS) is scheduled to be launched in 2019
on board the GEO-KOMPSAT (GEOstationary KOrea Multi-Purpose SATellite)-2B, contributing as
the Asian partner of the global geostationary constellation of air quality monitoring. To support this air
quality satellite mission, we perform the cross-verification of simulated GEMS ozone profile retrievals
based on the Optimal Estimation and ozonesonde measurements within the GEMS domain, covering
from 5°S (Indonesia) to 45°N (south of the Russian border) and from 75°E to 145°E. The comparison
between ozonesonde and GEMS shows a significant dependence on ozonesonde types. Ozonesonde
data measured by Modified Brewer-Master (MB-M) at Trivandrum and New Delhi show inconsistent
seasonal-variabilities in the tropospheric ozone, compared to latitudinally adjacent stations with Carbon
Iodine (CI) and Electrochemical Condensation Cell (ECC). CI ozonesonde measurements are biased
relative to ECC measurements by 2-4 DU; a better agreement with GEMS simulations is achieved with
ECC measurements. ECC ozone data at Hanoi, Kuala Lump, and Singapore show abnormally worse
agreements with simulated GEMS retrievals among ECC measurements. Therefore, ECC ozonesonde
measurements at Hong Kong, Pohang, Naha, Sapporo, and Tsukuba are finally identified as an optimal
reference. The accuracy of simulated GEMS retrievals is estimated to be ~ 5.0 % for both tropospheric
and stratospheric column ozone with the precision of 15 % and 5 %, which meet the GEMS ozone
requirements.



## 1. Introduction

The development of the geostationary ultraviolet (UV)/visible (VIS) spectrometers is highlighted toward a new paradigm in the field of the space-based air quality monitoring. It builds on the polar-orbiting instrument heritages for the last 40 years, which were initiated with the launch of a series of Total Ozone Mapping Spectrometer (TOMS) instruments since 1978 (Bhartia et al., 1996) and consolidated by Global Ozone Monitoring Experiment (GOME) (ESA, 1995), SCanning Imaging Absorption spectroMeter for Atmospheric CHartographY (SCIAMACHY) (Bovensmann et al., 1999), Ozone Monitoring Instrument (OMI) (Levelt et al, 2006), GOME/2 (EUMETSAT, 2006), Ozone Mapping Profiler Suite (OMPS) (Flynn et al., 2014), and TROPOspheric Monitoring Instrument (TROPOMI) (Veefkind et al., 2012). Three geostationary air quality monitoring missions, including the Geostationary Environmental Monitoring Spectrometer (GEMS) (Bak et al., 2013a) over East Asia, Tropospheric Emissions: Monitoring of pollution (TEMPO) (Chance et al, 2013; Zoogman et al., 2017) over North America, and Sentinal-4 (Ingmann et al., 2012) over Europe, are in progress to launch their instruments in the 2019-2022 time frame, which will provide unprecedented hourly measurements of aerosols and chemical pollutants at sub-urban scale spatial resolution ($\sim$ 10-50 km$^2$). These missions will constitute the global geostationary constellation of air quality monitoring.

GEMS will be launched in late 2019 on board the GeoKOMPSAT (Geostationary Korea Multi-Purpose Satellite) to measure $O_3$, $NO_2$, $SO_2$, $H_2CO$, $CHOCHO$, and aerosols in East Asia (Bak et al., 2013a). Tropospheric ozone is a key species to be monitored due to its critical role in controlling the air-quality as a primary component of photochemical smog, the self-cleansing capacity as a precursor of the hydroxyl radical, and in controlling the Earth's radiative balance as a greenhouse gas.

To support the development of the GEMS ozone profile algorithm, Bak et al. (2013a) demonstrated that the GEMS spectral coverage of 300-500 nm minimizes the loss in the sensitivity to tropospheric ozone despite the lack of most Hartley ozone absorption wavelengths shorter than 300 nm. They further indicated the acceptable quality of the simulated stratospheric ozone retrievals from 212 hPa to 3 hPa (40 km) through comparisons using Microwave Limb Sounder (MLS) measurements. As a consecutive work, this study evaluates simulated GEMS tropospheric ozone retrievals against ozonesonde observations. GEMS ozone retrievals are simulated using an optimal estimation based fitting algorithm from OMI radiances with the fitting window of 300-330 nm in the same way as Bak et al. (2013a). The validation effort is essential to ensuring the quality of GEMS ozone profile retrievals and to verifying the newly implemented ozone profile retrieval scheme. In-situ ozonesonde soundings have been considered to be the best reference, but should be carefully used due to its spatial and temporal



irregularities in instrument types, manufacturers, operating procedures, and correction strategies
(Deshler et al., 2017). Compared to TEMPO and Sentinel-4, validating GEMS ozone retrievals is
expected to be more challenging because of the much sparser distribution of stations and more irregular
characteristics of the ozonesonde dataset over the GEMS domain. Continuous balloon-borne
observations of ozone are only available from Pohang (129.23°E, 36.02°N) site in South Korea, but this
site have yet to be not been thoroughly validated. Therefore the quality assessment of the ozonesonde
data is required before we use this data for GEMS validation activity. Compared to ozonesondes,
satellite ozone data are less accurate, but more homogenous due to its single data processing for the
entire measurements from a single instrument. Therefore, abnormal deviations in satellite-ozonesonde
differences from neighboring stations might indicate problems at individual stations (Fioletov et al.
2008). For example, Bak et al. (2015) identified 27 homogenous stations among 35 global Brewer
stations available from the  World Ozone and Ultraviolet Radiation Data Centre (WOUDC) network
through comparisons with coincident OMI total ozone data. This study adopt this approach to select a
homogenous, consistent ozonesonde dataset among 10 stations available over the GEMS domain based
on the comparisons of the tropospheric ozone columns (TOC) between GEMS retrievals and
ozonesonde measurements, that is, simulated GEMS retrievals are used to verify the ozonesonde
observations. The simulated GEMS retrievals are ultimately evaluated against the ozonesonde dataset
identified as a true reference to demonstrate the reliability of our future GEMS ozone product. The
simulated GEMS retrievals and ozonesonde dataset are described in Sect. 2.1 and 2.2 with the
comparison methodology in Sect 2.3. Our results are discussed in Sect. 3 and summarized in Sect 4.
**2. Data and Methodology**
**2.1 Ozone Profile Retrievals**
The development of the GEMS ozone profile algorithm builds on heritages of the Smithsonian
Astrophysical Observatory (SAO) ozone profile algorithm which was originally developed for GOME
(Liu et al., 2005), continuously adapted for its successors such as OMI (Liu et al., 2010a), GOME/2
(Cai et al., 2012), and OMPS (Bak et al., 2017). In addition, the SAO algorithm will be implemented to
retrieve TEMPO ozone profiles (Chance et al., 2013; Zoogman et al., 2017). In this algorithm, the well-
known optimal estimation (OE) based iterative inversion is applied to estimate the best ozone
concentrations from simultaneously minimizing between measured and simulated backscattered UV
measurements constrained by measurement covariance matrix, and between retrieved values and its



climatological a priori values constrained by a priori covariance matrix (Rodgers, 2000). The impact of
a priori information on retrievals become important when measurement information is reduced due to
instrumental errors (e.g. straylight, dark-current, and read-out smear) or physically insufficient
sensitivities under extreme geophysical conditions (e.g. the reduced penetration of incoming UV
radiation into the lower troposphere at high solar zenith angles, blocked photon penetration below thick
clouds). The described OE-fitting solution $\hat{X}_{i+1}$ can be written, together with cost function $\chi^2$:

$$\hat{X}_{i+1} = \hat{X}_i + \left(K_i^T S_y^{-1} K_i + S_a^{-1}\right)^{-1}\{K_i^T S_y^{-1}[Y - R(\hat{X}_i)] - S_a^{-1}(\hat{X}_i - X_a)\} \quad (1)$$


$$\chi^2 = \left\|S_y^{-\frac{1}{2}} K_i(\hat{X}_{i+1} - \hat{X}_i) - [Y - R(\hat{X}_i)]\right\|_2^2 + \left\|S_a^{-\frac{1}{2}}(\hat{X}_{i+1} - X_a)\right\|_2^2 \quad (2)$$


Where $\hat{X}_{i+1}$ and $\hat{X}_i$ are current and previous state vectors with a priori vector, $X_a$ and its covariance
error matrix, $S_a$. $Y$ and $R(X)$ are measured and simulated radiance vectors, with measurement error
covariance matrix, $S_y$. $K$ is weighting function matrix ($\frac{dR(x)}{dx}$), describing the sensitivity of the forward
model to small perturbations of the state vector.
The ozone fitting window was determined toward maximizing the retrieval sensitivity to ozone
and minimizing that to measurement error: 289–307 nm and 326–339 nm for GOME, 270-309 nm and
312-330 nm for OMI, 289−307 nm and 325−340 nm for GOME/2, and 302.5-340 nm for OMPS. For
OMI, GOME and GOME/2, partial ozone columns are typically retrieved in 24 layers from the surface
to ~ 60 km. However, GEMS (300-500 nm) and OMPS (300-380 nm) do not cover much of the Hartley
ozone absorption wavelengths and hence the reliable profile information of ozone is limited at least
below ~ 40 km (Bak et al., 2013a).
Fig. 1 presents a schematic diagram of the ozone profile algorithm. With the input of satellite
measurements, the slit function is parameterized through cross-correlation between satellite irradiance
and high-resolution solar reference spectrum to be used for wavelength calibration and for high -
resolution cross section convolution (Sun et al., 2017; Bak et al., 2017); normalized Gaussian
distribution is assumed to derive analytic slit function for OMI. To remove the systematic errors
between measured and calculated radiances, "soft-calibration" is applied to measured radiances and
then the logarithm of sun-normalized radiances is calculated as a measurement vector (Liu et al., 2010a;
Cai et al., 2012; Bak et al., 2017). Measurement covariance matrix is constructed as a diagonal matrix
with each component taken from the square of the measurement errors as measurement errors are





assumed to be uncorrelated between wavelengths; for OMI the floor noise of 0.4 % (UV1) and 0.2 %
(UV2) is used because OMI measurement errors underestimate other kinds of random noise errors
caused by straylight, dark current, geophysical pseudo-random noise errors due to sub-pixel variability
and motion when taking a measurement, forward model parameter error (random part), and other
unknown errors (Huang et al., 2017). A priori ozone information is taken from tropopause-based (TB)
ozone profile climatology, which was developed for improving ozone profile retrievals in the upper
troposphere and lower stratosphere (Bak et al., 2013b). The Vector LInearized Discrete Ordinate
Radiative Transfer (VLIDORT) model (Spurr, 2006; 2008) is run to calculate the normalized radiance
and weighting function matrix for the atmosphere with Rayleigh scattering and trace-gas absorption
and with Lambertian reflection for both surface and cloud (Liu et al., 2010a). The ozone algorithm
iteratively estimates the best ozone profiles within the retrieval converges (typically 2-3 iterations),
together with other geophysical and calibration parameters (e.g., cloud fraction, albedo, BrO,
wavelength shifts, ring parameter, mean fitting scaling parameter) for a better fitting accuracy even
though some of the additional fitting parameters can reduce the degrees of freedom for signal of ozone.

**2.2 Ozonesonde measurements**

Ozonesondes are small, lightweight, and compact balloon-born instrument capable of measuring
profiles of ozone, pressure, temperature and humidity from the surface to balloon burst, usually near 35
km (4 hPa); ozone measurements are typically reported in the unit of partial pressure (mPa) with the
vertical resolution of ~ 100-150 m (WMO, 2014). Ozone soundings have been taken for more than 50
years since the 1960s. The accuracy of ozonesonde measurements has been reported as 5-10 % with the
precision of 3-5%, depending on the sensor type, manufacturer, solution concentrations, and operational
procedure (Smit et al., 2007; Thompson et al., 2007). The three types of instruments have been carried
on balloons, i.e. the Brewer-Mast (B-M), the electrochemical concentration cell (ECC), the carbon
iodine cell (CI). Each sounding is disposably operated and hence weekly launched for the long-term
operation.
Fig. 2 displays the locations of 10 ozonesonde sites focused on this study within the expected
GEMS domain bordered from 5°S (Indonesia) to 45°N (south of the Russian border) and from 75°E to
145°E. A summary of each ozonesonde site is present in Table 1. Most of measurements are collected
from the WOUDC network, except that Pohang soundings are provided from Korea Meteorological
Administration (KMA) and Kuala Lumpur and Hanoi measurements are from the Southern Hemisphere
Additional OZonesondes (SHADOZ) network. In South Korea, ECC sondes have been launched every



Wednesday since 1995 only at Pohang, without significant time gaps. There are three Japanese stations
(Naha, Tsukuba, and Sapporo) where the CI typed sensor was used and switched to the ECC-typed
sensor as of early 2009, and two Indian stations at New Delhi and Trivandrum using the Modified B-M
(MB-M) sensor. The rest of stations (Hanoi, Hong Kong, Kuala Lumpur and Singapore) uses only ECC.
Most stations employ an ECC ozone sensor, but inhomogeneities in ECC ozonesondes are strongly
addressed with respect to the preparation and correction procedures. There are two ECC sensor
manufactures; Science Pump Corporation (Model type: SPC-6A) and Environmental Science
Corporation (Model type: EN-SCI-Z/1Z/2Z). Since 2011 EN-SCI has been taken over by Droplet
Measurement Technologies (DMT) Inc. The Standard Sensing Solution has been recommended as
SST1.0 (1.0 % KI, full buffer) and SST 0.5 (2.0 % KI, no buffer) for the SPC and EN-SCI sondes,
respectively by the ASOPOS (Assessment for Standards on Operation Procedures for Ozone Sondes)
(Smit et al., 2012). Among ECC station, Pohang, Hong Kong, Japanese stations have applied the
standard sensing solution to all ECC observation with its one manufacture. In Singapore, the
ozonesonde manufacture was changed in late 2015 from EN-SCI to SPC, while SST 0.5 was switched
to SST 1.0 as of 2018. Two SHADOZ stations (Kuala lump, Hanoi) have applied the standard sensing
solution just since 2015. Hanoi changed sensing solution 4 times with two different ozonesonde
manufactures; Kula lump operated only with SPC 6A-SST 1.0 combination until 2014, but with four
different radiosonde manufactures. Therefore these SHADOZ dataset were reprocessed in Witte et al.
(2017) through the application of transfer functions between sensor and solution types to be
homogenized. The post-processing could be applied by data user to some WOUDC dataset given a
correction factor, which is the ratio of integrated ozonesonde column (appended with an estimated
residual ozone column above burst altitude) and total ozone measurements from co-located ground-
based and/or overpassing satellite instruments. The above-burst column ozone is estimated with a
constant ozone mixing ratio (CMR) assumption above the burst altitude (e.g., Japanese sites) (Morris
et al., 2013) or satellite derived stratospheric ozone climatology (e.g., Indian sites) (Rohtash et al., 2016).
No post-processing is given to Pohang, Hong Kong, and Singapore. Most stations made weekly or bi-
weekly regular observation, except for Indian stations with irregular periods of 0-4 per month and for
Singapore with monthly observation.

**2.3. Comparison Methodology**

The GEMS ozone profile algorithm is applied to OMI BUV measurements in 300-330 nm to
simulate GEMS ozone profile retrievals at coincident locations listed in Table 1. The coincidence



criteria between satellite and ozonesonde are: ±1.0º in both longitude and latitude and ±12 hours in time
and then the closest pixel is selected. The Aura satellite carrying OMI crosses the equator always at ~
1:45 pm LT and thereby OMI measurements are closely collocated within 3 hours to ozonesonde
soundings measured in afternoon (1-3 pm LS). Weekly based sonde measurements provide 48 ozone
profiles at maximum for a year; the number of collocation is on average 40 from 2004 October to 2008,
but reduced to ~ 20 recently due to the screened OMI measurements affected by the "row anomaly"
which is initially detected at two rows in 2007, seriously spread to other rows with time since January
2009 (Schenkeveld et al., 2017). As from July 2011 the row anomaly effect slowly extends up to ~ 50 %
of all rows. Correspondingly, the average collocation distance increases from 57.5 km to 66.6 km before
and after the occurrence of the row anomaly.

To increase the validation accuracy, the data screening is implemented to both ozonesonde

observation and satellite retrievals according to Huang et al (2017). For ozonesonde observation, we
screen ozonesondes with balloon-bursting altitudes exceeding 200 hPa, gaps greater than 3 km,
abnormally high concentration in the troposphere (> 80 DU), low concentration in the stratosphere
(<100 DU). Among WOUDC sites, Japanese and Indian dataset include a correction factor which is
derived to make a better agreement between integrated ozonesonde column and correlated reference
total ozone measurements as mentioned in Section 2.2; In Fig. 3, Japanese ozonesondes are compared
against GEMS simulations when a correction factor is applied or not to each CI and ECC measurements,
respectively. Morris et al. (2013) recommended to restrict the application of this correction factor to the
stratospheric portion of the CI ozonesonde profiles due to errors in the above-burst column ozone. Our
comparison results illustrate that applying the correction factor reduces the vertical fluctuation of mean
biased in ozone profile differences with insignificant impact on their standard deviations. Therefore we
decide to apply this correction factor to the sonde profiles if this factor ranges from 0.85 to 1.15. Because
of a lack of retrieval sensitivity to ozone below clouds and lower tropospheric ozone under extreme
viewing condition, satellite retrievals are limited to cloud fraction less than 0.5, SZAs less than 60°, and
fitting RMS (i.e., root mean square of fitting residuals relative to measurement errors) less than 3.

Due to the different units of ozone amount between satellite and ozonesonde, we convert

ozonesonde-measured partial pressure ozone values (mPa) to partial column ozone (DU) at 24 retrieval
grids of satellite for the altitude range from surface to the balloon-bursting altitudes. Ozonesonde
measurements are obtained at a rate of a few seconds and then typically averaged into altitude
increments of 100 meters, whereas retrieved ozone profiles from nadir BUV satellite measurements
have much coarser vertical resolution of 10-14 km in the troposphere and 7-11 km in the troposphere
based on OMI retrievals. Consequently, satellite observation captures the smoothed structures of
ozonesonde soundings, especially in the tropopause, where a sharp vertical transition of ozone within 1



km is observed, and in the boundary layer due to the insufficient penetration of photon. Satellite
retrievals unavoidably have an error compound due to its limited vertical resolution, which is named
"smoothing error" in the OE based retrievals (Rodgers, 2000). It could be useful to eliminate the effect
of smoothing errors on differences between satellite and sonde to better characterize other error sources
in the comparison (Liu et al., 2010a). For this reason, satellite data have been compared to smoothed
ozonesonde measurements into the satellite vertical resolution together with original sonde soundings
(Liu et al., 2010b; Bak et al., 2013b; Huang et al., 2017). The smoothing approach is following as

$\hat{x}_{sonde} = A \cdot x_{sonde} + x_a(1 - A)$                                (3)

$x_{sonde}$ : High-resolution ozonesonde profile

$\hat{x}_{sonde}$ : Convolved ozonesonde profile into satellite vertical resolution

A       : Satellite averaging kernel

$x_a$     : A priori ozone profile


In order to define tropospheric columns, both satellite retrievals and ozonesonde measurements

are vertically integrated from the surface to the tropopause taken from daily National Centers for
Environmental Prediction (NCEP) final (FNL) Operational Global analysis data
(http://rda.ucar.edu/datasets/ds083.2/). To account for the effect of surface height differences on
comparison, ozone amount of satellite data below the surface height of ozonesonde is added to
tropospheric columns of ozonesonde measurements and vice versa.

**3. Results and Discussions**

**3.1 Comparison at individual stations**

Witte et al. (2018) recently compared seven SHADOZ station ozonesonde records, including

Hanoi and Kuala Lumpur in the GEMS domain, with total ozone and stratospheric ozone profiles
measured by space-borne nadir and limb viewing instruments, respectively. In this comparison, Hanoi
station shows comparable or better agreement with the satellite dataset when compared to other sites.
Morris et al. (2013) and Rohtash et al. (2016) thoroughly evaluated ozonesonde dataset over Japanese
and Indian sites, respectively, but they did not address their measurement accuracy with respect to those
at other stations. Validation of GOME TOC by Liu et al. (2006) showed relatively larger biases at
Japanese CI stations and validation of OMI TOC by Huang et al. (2017) showed both larger biases and



standard deviations at the India MB-M sites. In South Korea, regular ozonesonde measurements are
taken only from Pohang, but these measurements have been insufficiently evaluated; only the
stratospheric parts of these measurements were quantitatively assessed against satellite solar occultation
measurements by Halogen Occultation Experiment (HALOE) from 1995 to 2004 in Hwang et al. (2006),
but only 26 pairs were compared despite its coarse coincident criteria (48 hours in time, $\pm4.5^\circ$ in
latitude, $\pm9^\circ$ in longitude). Therefore, it is important to perform the quality assessment of ozonesonde
measurements to identify the reliable reference dataset for GEMS ozone profile validation
For this purpose, we illustrate tropospheric ozone columns (TOC) as a function of time and
individual stations listed in Table 1, measured with three different types of ozonesonde instruments and
retrieved with GEMS simulations (Fig. 4), respectively. The goal of this comparison is to identify any
abnormal deviation of ozonesonde measurements relative to satellite retrievals, so we exclude the
impact of the different vertical resolutions between instruments and satellite retrievals on this
comparison by convolving ozonesonde data with satellite averaging kernels. At mid-latitude sites
(Pohang, Sapporo, and Tsukuba) both ozonesonde and satellite retrievals show the distinct seasonal
TOC variations with the amplitude of ~ 35-40 DU. Extratropical sites (Naha, Hong Kong, and Hanoi)
show less seasonal variations of 30 to 50 DU, whereas fairly constant concentrations are observed at
Kuala Lumpur and Singapore in tropics. Both ozonesonde observations and satellite retrievals illustrate
similar seasonal variabilities at these locations. At New Delhi and Trivandrum, on the other hand, MB-
M ozonesonde measurements abnormally deviate from 10 DU to 50 DU compared to the corresponding
satellite retrievals and latitudinally neighboring ozonesondes.
In Fig. 5 time dependent errors in differences of TOC between ozonesonde and satellite retrievals
are evaluated with the corresponding comparison statistics in Table 2. Satellite retrievals show strong
correlation of ~ 0.8 or much larger with ozonesonde measurements at Pohang, Hong Kong, and three
Japan stations, and with less correlation of ~ 0.5 at other SHADOZ stations in the tropics. However,
Indian stations show poor correlation of 0.24. Mean biases and its standard deviations are much smaller
at stations where a strong correlation is observed; they are ~1 DU $\pm$~ 4DU at most ECC stations,
but deviated to ~ 4 DU $\pm$~ 10 DU at MB-M stations. In conclusion, we should exclude ozonesonde
observations measured by MB-M to remove irregularities in a reference dataset for validating both
GEMS simulated retrievals in this study and GEMS actual retrievals in future study. Moreover, time
series of ozonesonde and satellite observations show a significant transition at three Japan stations as
of late 2008 and early 2009 when the ozonesonde instrument was switched from CI to ECC. This
transition could be affected by space-born instrument degradation, but the impact of balloon-born
instrument change on them is predominant based on less time-dependent degradation pattern at



latitudinally neighboring stations during this period. CI ozonesonde noticeably underestimates
atmospheric ozone by 2-3 DU compared to ECC and thereby GEMS TOC biases relative to CI
measurements, are estimated as - 2 to - 5 DU but these biases are reduced to < 1.5 DU when compared
with ECC. Therefore, we decide to exclude these CI ozonesonde observations for evaluating GEMS
simulated retrievals. Compared to other ECC stations, Hanoi station often changed sensing solution
concentrations and pH buffers (Table 1) and hence might cause the irregularities due to remaining errors
even though transfer functions were applied to ozonesonde measurements to account for errors due to
the different sensing solution (Witte et al., 2017). This fact might affect the relatively worse performance
compared to latitudually adjacent station, Hong Kong, where the 1.0 % KI buffered sensing solution
(SST 1.0) to ECC/SPC sensors have been consistently applied.
Fig. 6 compares differences of ozone profiles between ECC ozonesondes and GEMS simulated
retrievals at each station. Among ECC ozonesondes, Singapore ozonesondes are in the worst agreement
with satellite retrievals in both terms of mean biases and standard deviations, which could be explained
by the discrepancy of collocation time. Sonde observations at Japan, Pohang, Hong Kong, and Hanoi
stations, where balloons were launched in afternoon (~ 12-15 LST), are collocated within ~ 1-2 h to
OMI that passes the equator at 13:45 LST and then reaches the pole within 25 min, whereas the time
discrepancy increases to 7 h at Singapore where ozonesondes are launched in the early morning.
Photochemical ozone concentrations are typically denser in the afternoon than in the morning and hence
ozonesonde measurements at Singapore are negatively biased relative to afternoon satellite
measurements. For the reason mentioned above, the discrepancy in the observation time could impact
on this comparison at Kuala Lump, where sondes were mostly launched in the late morning, 2-3 hours
prior to the OMI passing time and thereby ozonesonde measurements tend to be negatively biased.
These indicate that diurnal variations of the tropospheric ozone are visible in oznesonde measurements,
emphasizing on hourly geostationary ozone measurements. The comparison results could be
characterized with latitudes. In the mid-latitude, noticeable disagreements are commonly addressed in
tropopause region where mean biases/standard deviations are ~10 %/~15% larger than those in the
lower troposphere. In the extra-tropics (Hong Kong, Naha), consistent differences of - a few % are
shown over the entire altitude with standard deviations of 15 % or less below the tropopause (~ 15 km).
Hanoi and Kuala Lump show significantly larger biases/standard deviations compared to other ECC
stations. At Hanoi inconsistencies of solution concentrations and pH buffers might influence on this
instability. At Kuala Lump the inconsistencies of observation times might be one of the reasons,
considering its standard deviations of ~100 min, but mostly less than 30 min at other stations. Therefore,
we strictly screen out Singapore, Kuala Lump, and Hanoi, together with all M-BM measurements at
Indian stations and CI measurements at Japanese stations to improve the validation accuracy of GEMS



simulated retrievals in next section. Eventually, stations, where the standard procedures for preparing
and operating ECC sondes are consistently maintained, are accepted as an optimal reference in this
work.

**4.2 Evaluation of GEMS simulated ozone profile retrievals**

The GEMS simulated retrievals are assessed against ECC ozonesonde soundings at five stations
(Hong Kong, Pohang, Tsukuba, Sapporo, and Naha) identified as a good reference in the previous
section. The comparison statistics include mean bias and standard deviation in the absolute/relative
differences, correlation coefficient, the linear regression results (slope (a), intercept (b), error); the error
of the linear regression is defined as $\frac{1}{n}\sqrt{\sum_i^n (y_{GEMS} - y_{fit})^2}$, $y_{fit} = a \cdot y_{sonde} + b$. In Fig. 7, GEMS
simulated retrievals are plotted as a function of ozonesonde with and without the vertical resolution
smoothing, respectively, for the stratospheric and tropospheric columns. GEMS simulations
underestimate the tropospheric ozone by $\sim 2.27 \pm 5.94$ DU and overestimate the stratospheric ozone
by $\sim 9.35 \pm 8.07$ DU relative to high-resolution ozonesonde observations. This comparison
demonstrates a good correlation coefficient of 0.84 and 0.99 for troposphere and stratosphere,
respectively. This agreement is degraded if the rejected ECC sondes (Kuala Lump, Hanoi, and
Singapore) are included; for example, the slope decreases from 0.68 to 0.64 while the RMSE increases
6.35 and 6.76 DU for TOC comparison. Smoothing ozonesonde soundings into GEMS vertical
resolution improves the comparison results, especially for the tropospheric ozone columns; standard
deviations are reduced by $\sim 5$ % with mean biases of less than 1 DU. Similar assessments are performed
for OMI standard ozone profiles based on the KNMI OE algorithm (Kroon et al., 2011) hereafter
referred to as OMO3PR (KNMI) in Fig. 8 and the research product based on the SAO algorithm (Liu
et al., 2010) hereafter referred to as OMPROFOZ (SAO) in Fig. 9, respectively. It implies that GEMS
gives the good information on SOCs comparable to both OMI KNMI and SAO products in spite of
excluding most of Hartley ozone band in GEMS retrievals. Furthermore, a better agreement of GEMS
TOCs with ozonesonde is found than others due to different implementation details. As mentioned in
2.1., GEMS algorithm is developed based on the heritages of the SAO ozone profile algorithm with
several modifications. There are two main modifications: a priori ozone climatology was replaced with
a tropopause-based ozone profile climatology to better represent the ozone variability in the tropopause.
Irradiance spectra used to normalize radiance spectra and characterize instrument line shapes are
prepared by taking 31-day moving average instead of climatological average to take into account for
time-dependent instrument degradations. These modifications reduce somewhat spreads in deviations





of satellite retrievals from sondes, especially in TCO comparison. KNMI retrievals systematically
overestimate the tropospheric ozone by ~ 6 DU (Fig. 9.c), which corresponds to the positive biases of
2-4 % in the integrated total columns of KNMI profiles relative to Brewer observations (Bak et al.,
2015). As mentioned in Bak et al. (2015), the systematic biases in ozone retrievals are less visible in
SAO-based retrievals (GEMS simulation, OMPROFOZ) as systematic components of measured spectra
are taken into account for using an empirical correction called "soft calibration".

**4. Summary**

We simulate GEMS ozone profile retrievals from OMI BUV radiances in the range of 300-330 nm
using the optimal estimation based fitting during the period of 2005-2015 to ensure the performance of
the algorithm against coincident ozonesonde observations. There are 10 ozonesonde sites over the
GEMS domain from WOUDC, SHADOZ and KMA archives. This paper gives an overview of these
ozonesonde observation systems to address inhomogeneities in preparation, operation, and correction
procedures which cause discontinuities in individual long-term records or in adjoint stations.
Comparisons between GEMS TOC retrievals and ozonesondes illustrate a noticeable dependence on
the instrument type. Indian ozonesonde soundings measured by MB-M show severe deviations in
seasonal time series of TOC compared to coherent GEMS simulations and neighboring ozonesondes.
At Japanese stations, CI ozonesondes underestimate ECC ozonesonde by 2 DU or more and a better
agreement with GEMS simulations is found when ECC measurements are compared. Therefore, only
ECC ozonesonde measurements are first selected as a reference, in order to ensure a consistent,
homogeneous dataset. Furthermore, ECC measurements at Singapore, Kuala Lump, and Hanoi are
excluded. At Singapore and Kuala Lump, observations were performed in the morning and thereby
inconsistent with GEMS retrievals simulated at OMI overpass time in the afternoon. In addition,
observation time for Kuala Lump is inconsistent itself compared to other stations; its standard deviation
is ~ 100 min, but for other ECC stations less than 30 min. At Hanoi the combinations of sensing solution
concentrations and pH buffers changed 4 times during the period of 2005 through 2015. Therefore,
GEMS and ozonsonde comparisons show larger biases/standard deviations at these stations. Pohang
station is unique in South Korea where ECC ozonesondes have been regularly and consistently launched
without gap since 1995; the standard 1% KI full buffered sensing solution has been consistently applied
to ozone sensors manufactured by SPC (6A model). Evaluation of Pohang ozonesondes against GEMS
simulations demonstrates its high level reliability, which is comparable to latitudually adjacent Japanese
ECC measurements at Tsukuba and Sapporo. Reasonable agreement with GEMS retrievals is s similarly
shown at Latitudually adjacent Naha and Hong Kong stations. Finally, we establish that the comparison



statistics of GEMS simulated retrievals and optimal reference dataset is -2.27 (4.92) $\pm$ 5.94 (14.86)
DU (%) with R = 0.84 for the tropospheric columns and 9.35 (5.09) $\pm$ 8.07 (4.60) DU (%) with R=0.99
for the stratospheric columns. This estimated accuracy and precision is comparable to OMI products
for the stratospheric ozone column and even better for the tropospheric ozone column due to improved
implementations. Our future study aims to achieve this quality level from actual GEMS ozone profile
product.

**Acknowledgement**
The ozonesonde data used in this study were obtained though the WOUDC, SHADOZ and KMA
archives. We also acknowledge the OMI science team for providing their satellite data. Research at the
Smithsonian Astrophysical Observatory was funded by NASA and the Smithsonian Institution.
Research at Pusan National University was supported by Basic Science Research Program through the
National Research Foundation of Korea (NRF) funded by the Ministry of Education
(2016R1D1A1B01016565). This work was also supported by the Korea Ministry of Environment
(MOE) as the Public Technology Program based on Environmental Policy (2017000160001).

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


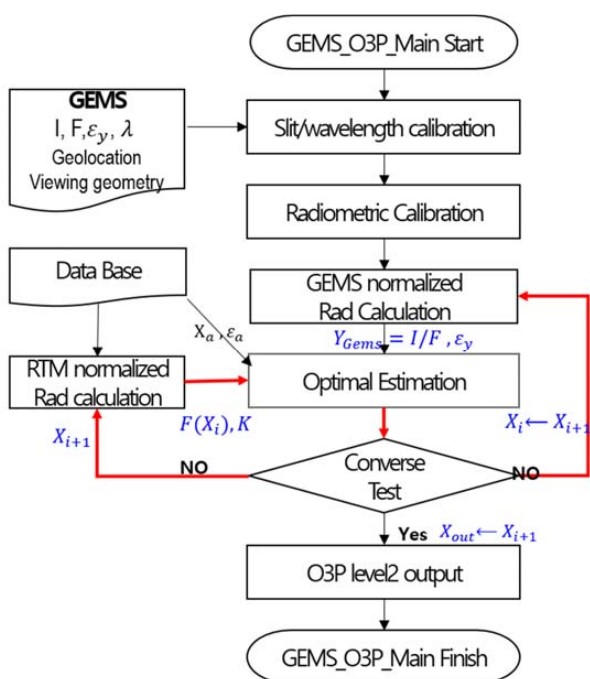

**Figure 1.** Flow Chart of the GEMS ozone profile retrieval algorithm.





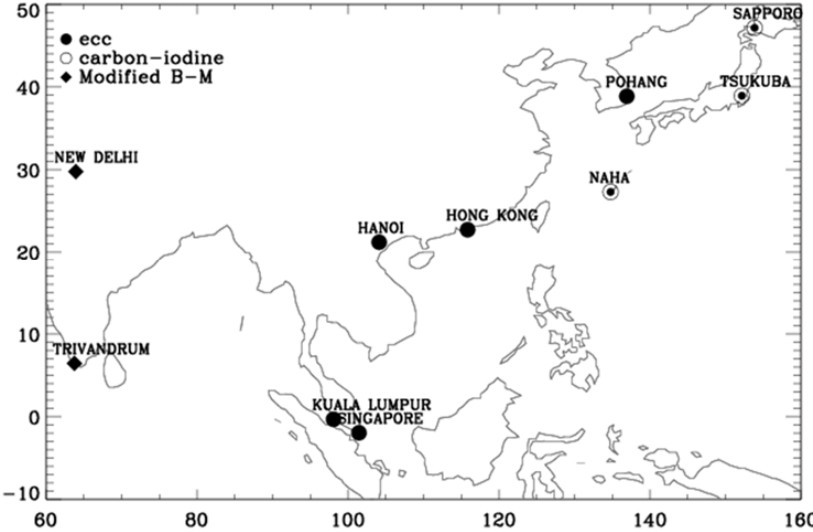

**Figure 2.** Geographic locations of the ozonesonde stations available since 2005 over the GEMS observation domain.





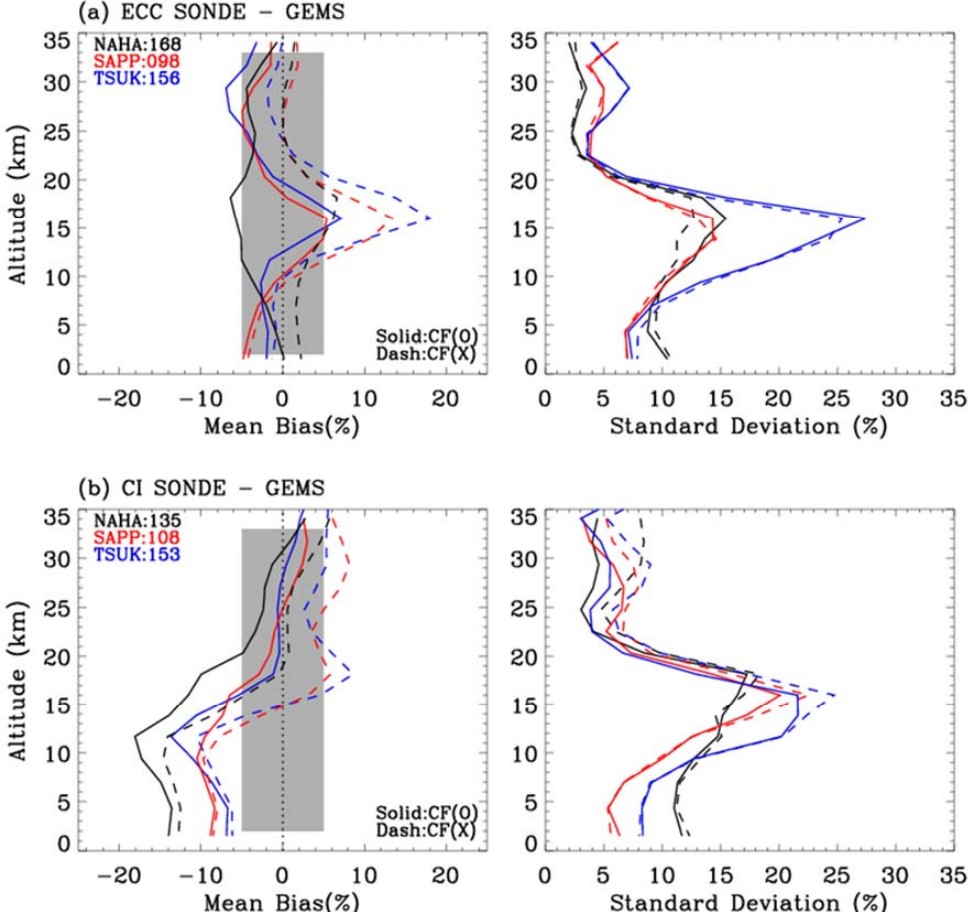

**Figure 3.** Effect of applying a correction factor to (a) ECC and (b) CI ozonesonde measurements, respectively on comparisons with simulated GEMS ozone profile retrievals.



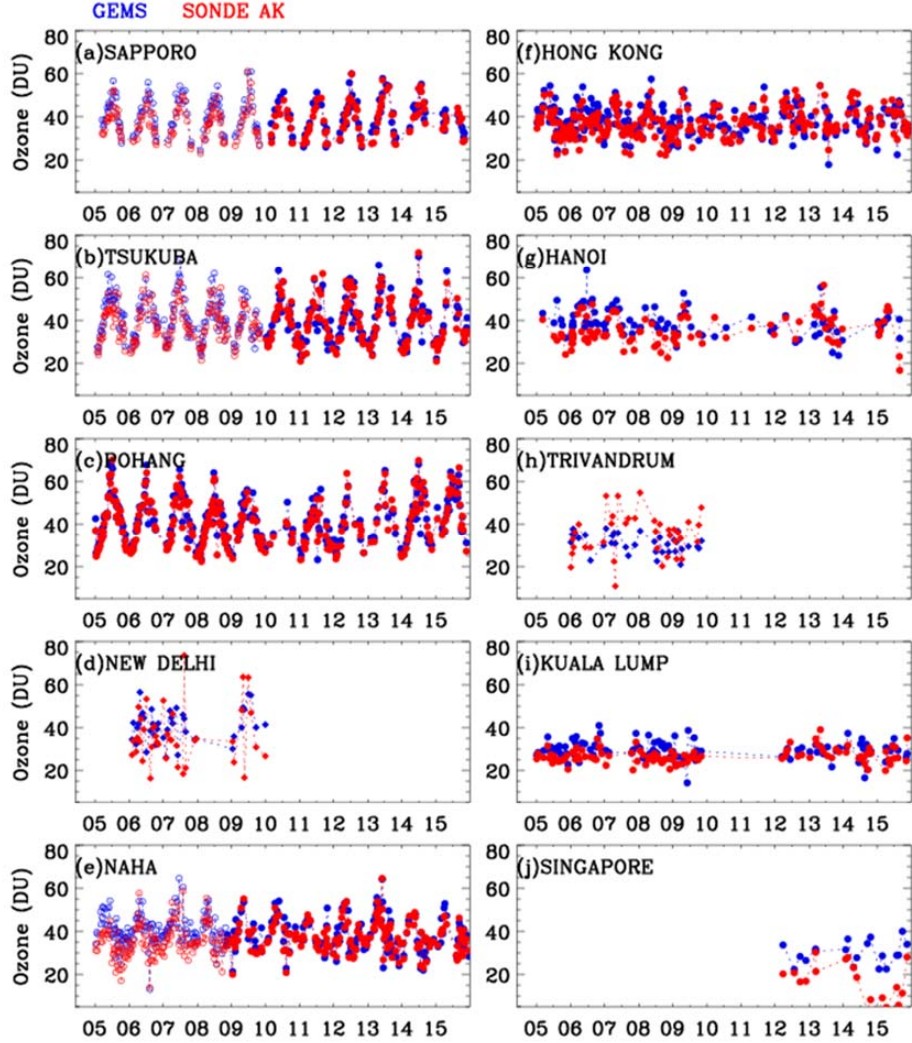

**Figure 4**. Time series of tropospheric ozone columns (DU) of GEMS simulated ozone profile retrievals (blue) and ozonesonde measurements convolved with GEMS averaging kernels (red) from 2005 to 2015 at 10 stations listed in Table 1.



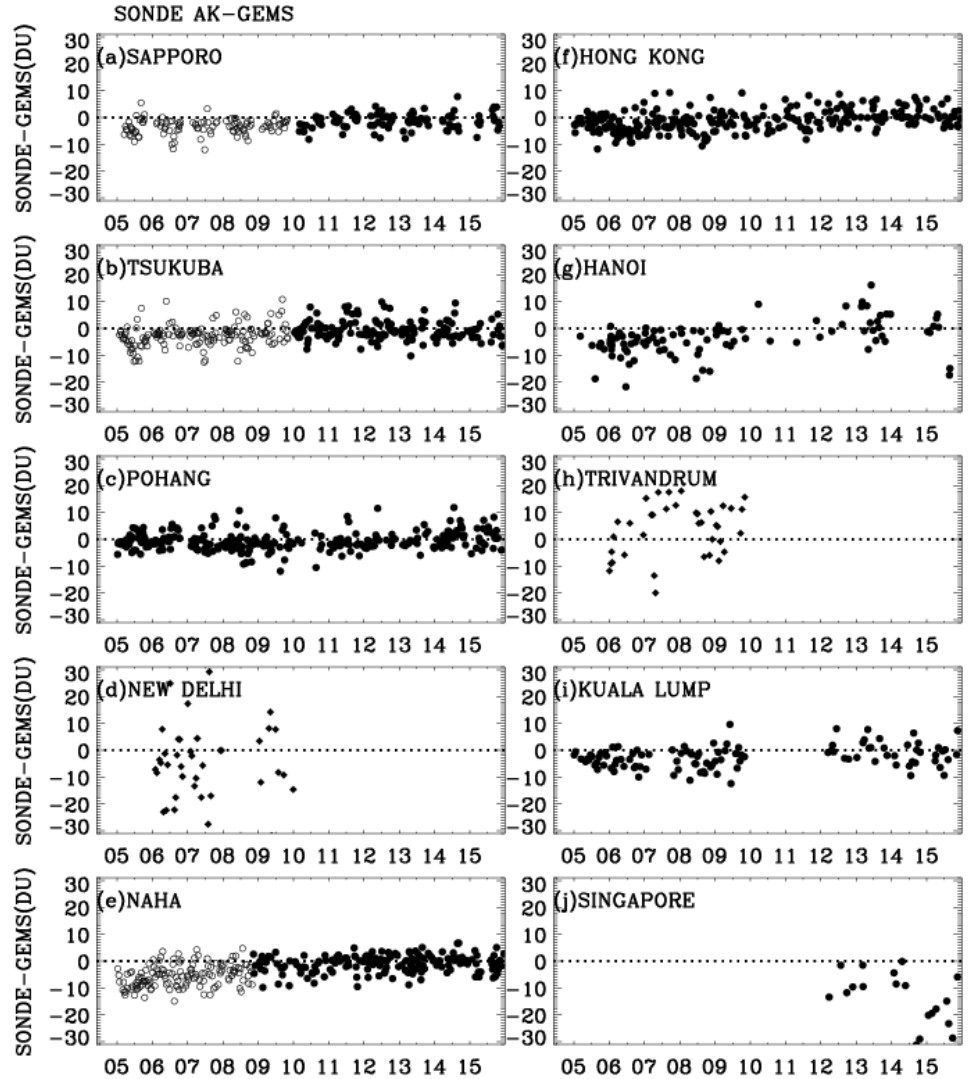

**Figure 5**. Same as Figure 4, but for absolute differences of tropospheric ozone columns (DU) between ozonesonde measurements and GEMS simulated retrievals.





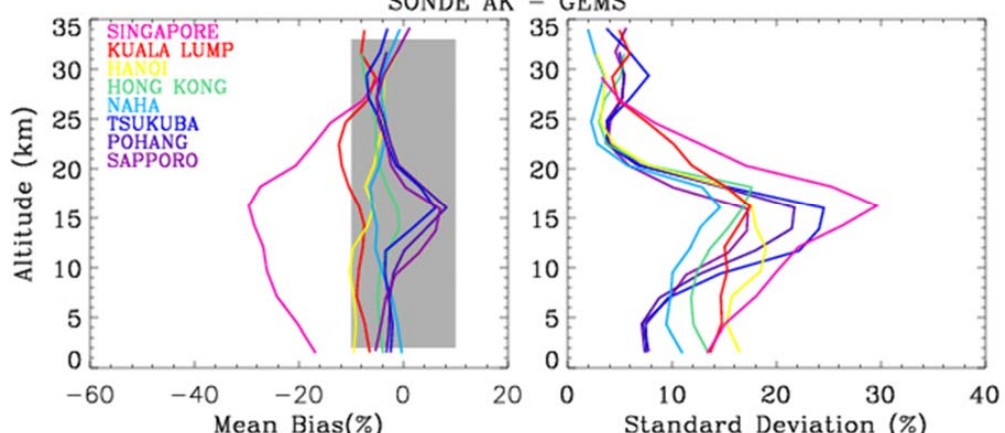

**Figure 6**. Mean biases and 1σ standard deviations of the differences between ozonesonde convolved
with GEMS averaging kernels and GEMS simulated ozone retrievals as a function of GEMS layers, at
ECC ozonesonde stations. The relative difference is defined as 2 (SONDE AK – GEMS) X100 %/ (A
priori).





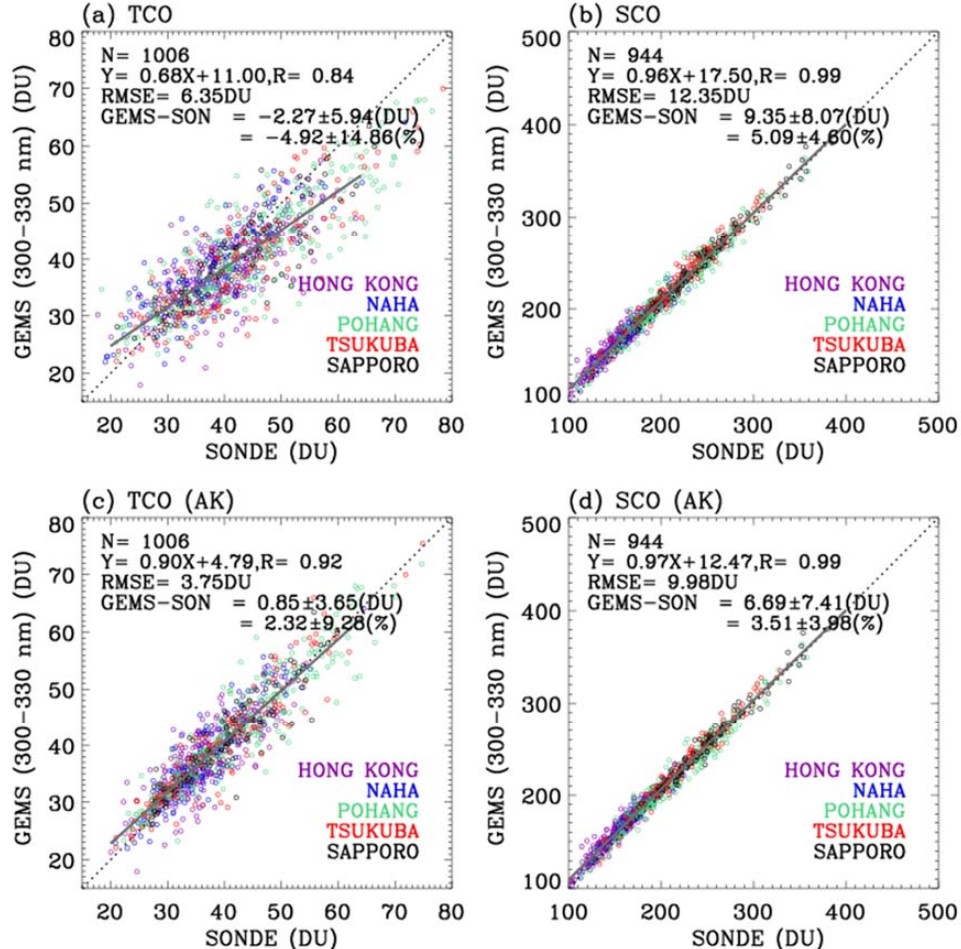

**Figure 7**. Upper: Scatter plots of GEMS vs. ozonesonde for tropospheric and stratospheric ozone columns, respectively. Lower panel is same as Upper one, except that ozonesonde measurements are convolved with GEMS averaging kernels. A linear fit between them is shown in red, with the 1:1 lines (dotted lines). The legends show the number of data points (N), the slope and intercept of a linear regression, and correlation coefficient (r), with mean biases and 1σ standard deviations for absolute (DU) and relative differences (%), respectively. Note that we use 5 stations identified as a good reference among 10 stations listed in Table 1 in this comparison.





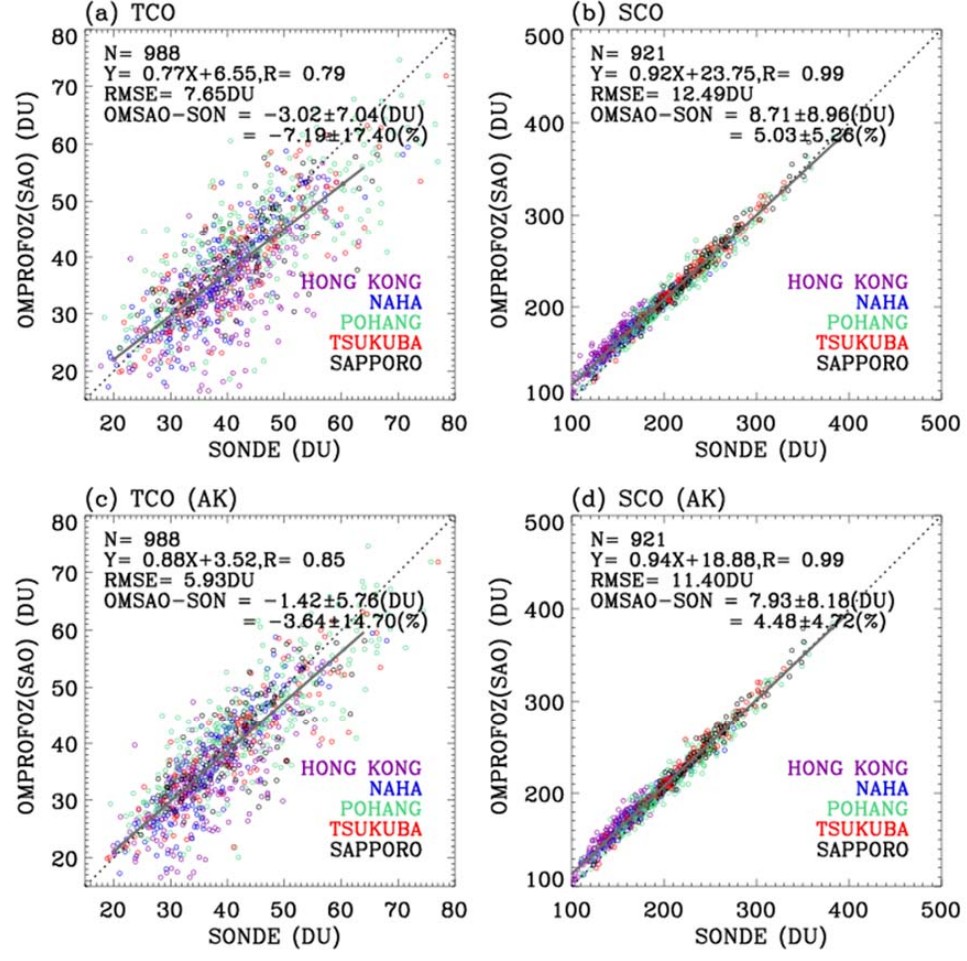

**Figure 8**. Same as Fig. 8, but for validating OMI research ozone profile (OMPROFOZ) produced by the SAO optimal estimation based algorithm.



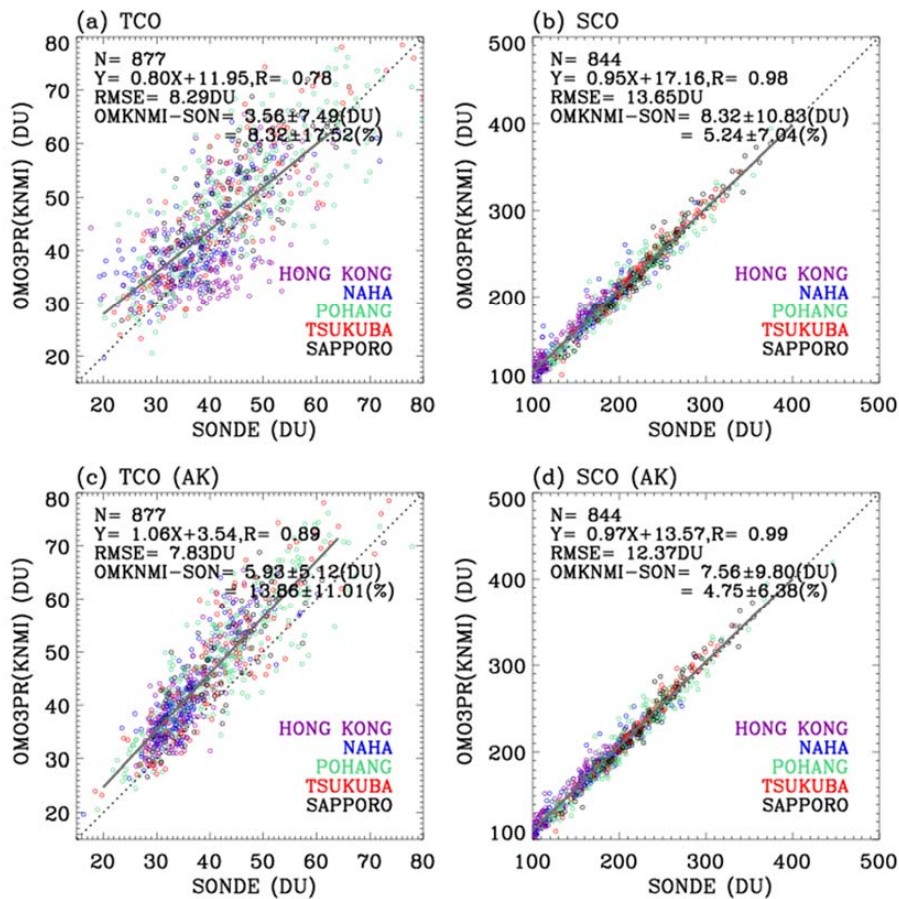

**Figure 9**. Same as Fig. 7, but for validating OMI standard ozone profiles (OMO3PR) produced by the KNMI optimal estimation based algorithm.



**Table1.** List of ozonesonde stations.

| Station[a] | Lon(°), Lat(°) | Altitude (m) | Observation Time[b] | | Instrument Type[c] | ECC-SST[d] | Post Correction |
|---|---|---|---|---|---|---|---|
| Singapore | 103.9, 1.3 | 40 | 07:30-08:00 (9) | Jan 12 - Sep 15 | ECC/EN-SCI Z | | No correction |
| | | | | Nov15 - Dec15 | ECC/SPC 6A | SST0.5 | |
| Kuala lump | 101.7, 2.7 | 20 | 9:30-15:00 (104) | Jan 13 - Dec14 | ECC/SPC 6A | SST1.0 | Transfer function |
| | | | | Jan 15 - Dec15 | ECC/EN-SCI Z | SST0.5 | |
| Trivandrum | 77.0, 8.5 | 60 | 14:00-14:30 (34) | Jan 06 - Dec11 | MB-M | | Correction factor |
| Hanoi | 105.8, 21.0 | 10 | 12:00-14:00 (42) | Jan 05 - Apr 06 | ECC/EN-SCI 1Z | SST2.0 | Transfer function |
| | | | | Apr06 - Dec 07 | ECC/EN-SCI 2Z | SST2.0 | |
| | | | | Jan 08 - May 09 | ECC/EN-SCI 2Z | SST1.0 | |
| | | | | Jun 09 - Dec 09 | ECC/SPC 6A | SST1.0 | |
| | | | | Feb 10 - Dec 11 | ECC/EN-SCI Z | SST1.0 | |
| | | | | Feb 12 - Dec 13 | ECC/EN-SCI Z | SST2.0 | |
| | | | | Jan 15 - Dec 15 | ECC/EN-SCI Z | SST0.5 | |
| Hong Kong | 114.1, 22.3 | 70 | 13:00-14:30 (11) | Jan 05 - Dec 15 | ECC/SPC 6A | SST1.0 | No correction |
| Naha | 127.7, 26.2 | 30 | 14:30-15:00 (06) | Jan 05 - Oct 08 | CI/ KC-96 | | Correction factor |
| | | | | Nov 09 - Dec 15 | ECC/EN-SCI 1Z | SST0.5 | |
| New Delhi | 77.1, 28.3 | 270 | 11:00-14:30 (69) | Feb 06 - Dec11 | MB-M | | Correction factor |
| Pohang | 129.2, 36.0 | 40 | 13:30-15:30 (24) | Jan 05 - Dec 15 | ECC/SPC 6A | SST1.0 | No correction |
| Tsukuba | 140.1, 36.1 | 330 | 14:30-15:00 (08) | Jan 05 - Nov 09 | CI/ KC-96 | | Correction factor |
| | | | | Dec 09 - Dec 15 | ECC/EN-SCI 1Z | SST0.5 | |
| Sapporo | 141.3, 43.1 | 30 | 14:30-15:00 (06) | Jan 05 - Nov 09 | CI/ KC-96 | | Correction factor |
| | | | | Dec 09 - Dec 15 | ECC/EN-SCI 1Z | SST0.5 | |

[a] Data are downloaded from WOUDC (http://woudc.org) data archive, except for Kuala lump and Hanoi, which are from SHADOZ (https://tropo.gsfc.nasa.gov/shadoz/) network, and Pohang, which are from Korea Meteorological Administration (KMA).
[b] The range of the observation time (LT) with 1 σ standard deviations of them (min) in the parentheses.
[c] Ozonesonde sensor type (ECC: Electrochemical Condensation Cell, CI: Carbon iodine cell Japanese sonde, MB-M: Modified Brewer-Mast Indian sonde). ECC sensors manufactured by either ECC sensor manufactures; Science Pump Corporation (Model type: SPC-6A) and Environmental Science cooperation (Model type EN-SCI-Z/1Z/2Z).
[d] Potassium Iodide (KI) cathode sensing solution type (SST) implemented in ECC ozone sensors: SST 0.5 (0.5 % KI, half buffer), SST 1.0 (1.0 % KI, full buffer), and SST 2.0 (2.0 % KI, no buffer). Singapore station changed it to SST 1.0 % as of 2018.



Table 2. Comparison Statistics (Mean Bias in DU, 1s Standard Deviation in DU, and Correlation Coefficient) between GEMS simulated Tropospheric Ozone Column and Ozonesonde Measurements convolved with GEMS averaging kernels.

| Station | Collocation Time difference | Type | Data Period (Year) | SONDE AK – GEMS | | |
|---|---|---|---|---|---|---|
| | | | | # | Mean Bias + 1σ | R |
| Singapore | 6:44 | ECC | 12-15 | 20 | -13.67 ± 9.61 | 0.17 |
| Kuala lump | 2:29 | ECC | 05-15 | 106 | -2.54 ± 4.13 | 0.44 |
| Trivandrum | 1:46 | MB-M | 06-11 | 37 | 3.55 ± 9.75 | 0.24 |
| Hanoi | 0:32 | ECC | 05-15 | 100 | -3.82 ± 6.03 | 0.52 |
| Hong Kong | 0:27 | ECC | 05-15 | 259 | -1.19 ± 3.91 | 0.82 |
| Naha | 0:47 | CI | 05-08 | 135 | -5.48 ± 4.07 | 0.85 |
| | | ECC | 08-15 | 166 | -0.94 ± 3.22 | 0.91 |
| New Delhi | 1:46 | MB-M | 06-11 | 39 | -4.57 ± 13.36 | 0.24 |
| Pohang | 0:54 | ECC | 05-15 | 281 | -0.75 ± 3.13 | 0.95 |
| Tsukuba | 1:56 | CI | 05-09 | 151 | -2.98 ± 3.76 | 0.91 |
| | | ECC | 09-15 | 154 | -0.65 ± 3.53 | 0.94 |
| Sapporo | 2:18 | CI | 05-09 | 107 | -3.43 ± 2.56 | 0.94 |
| | | ECC | 09-15 | 95 | -1.37 ± 2.79 | 0.93 |