# Peer review of "Cross-evaluation of GEMS tropospheric ozone"

_Atmospheric Measurement Techniques, 2019_

## Referee Comment (RC1) · Anonymous Referee #1 · 13 Mar 2019

Main comments

1. This paper is about the testing of the GEMS retrieval algorithm on OMI radiances, and the subsequent comparison with ozonesondes. This is interesting and worthwhile study in the run-up to the launch of GEMS.

2. I see the paper as having two main purposes with respect to GEMS. First, it is an exercise of the retrieval algorithm on real OMI data. The fact that this is successful gives confidence that the retrieval is ready to receive the first GEMS data after launch. However, quantitative verification of the retrieval performance is harder, and the discussion of the GEMS retrieval algorithm performance on OMI radiances against sondes, com-

pared with the OMI-algorithm retrievals against the same sondes should be expanded. The second purpose is to identify those ozonesonde measurements that might be good for GEMS validation, in as much as the work here suggests that they are useful or not for OMI validation.

3. If there is a cross-verification here, it is really about OMI validation between OMI and the radiosondes. The fact that the GEMS algorithm is used to process the OMI radiances does not change this, especially with comparisons that should adequately account for how a priori profile and smoothing error assumptions differ between the GEMS and usual OMI algorithms. As such, the title of the manuscript does not clearly describe what is done in the paper, and I suggest that the authors modify the title to better reflect the above two goals.

4. Section 2.1 describes the retrieval algorithm applied to the OMI radiances. A discussion should be included as to how the retrieval algorithm characteristics are expected to change for the GEMS radiances.

5. Section 3.1. The discussion of the differences between satellite/sonde agreement at the different sites is interesting. In addition to the differences between the sonde characteristics and reliabilities, one might expect greater standard deviations at sites that are polluted and/or show greater variability in ozone loadings due to meteorology. Some further discussion would be useful about the chemical-transport environment before eliminating sites from potential GEMS validation based on instrumentation/experimental method arguments alone. It would also help this reader if the current dense text were broken up into descriptions of the various reasons for good/bad agreement.

6. Section 4.2 and Fig. 7. The impact on correlation of smoothing or not smoothing the sonde profiles might be dependent on how close the GEMS retrieval a priori profile is to the sonde "truth". How do these compare and how do they vary between locations of good and poor comparison? How far does a priori profile go toward explaining the

bias?

7. Section 4.2 should be split into another sub section at Line 354 that starts the discussion of the evaluation of the GEMS algorithm against the OMI algorithm. This should be presented as one of the main results sections of this paper: a quantitative evaluation of the GEMS algorithm against other widely used algorithms based on the same OMI radiances.

8. Given the previous discussion in the paper of the various limitations of some of the sondes, it might additionally be useful to directly compare the results of the different retrieval algorithms and explain differences in results on the basis of different features of the algorithms. This would help make the case that the GEMS algorithm is performing as expected.

9. The paper requires careful, and extensive, editing for English usage, and cut-paste typos, e.g. line 75, that should have been corrected before manuscript submission.

Minor comments

• Several times "GEMS" measurements are described. The word "simulated" should be added each time to avoid confusion.

• Line 83: consistent perhaps, but not homogeneous as the authors point out in the text above.

• Line 100: Rodgers method is more correctly described as maximum a postiori.

• Line 105: Instrument errors certainly, but also instrument design sensitivity.

• Line 106: Common geophysical conditions can reduce sensitivity, not just extreme.

• Line 123: Information may be limited but is a goal of the GEMS mission. This should be clarified.

• Line 157: Any more recent references to new measurement technique and instrumentation? • Line 200: How do these coincidence criteria for OMI and the sondes affect the results? What is the expected variation within the time and space windows? What is the representativeness uncertainty? How do these results here with the OMI comparison inform on the expected GEMS comparisons with hourly measurements at ~7km resolution?

• Line 231: Is "troposphere" written where is should be stratosphere?
* * *

---

## Author Comment (AC1) · 30 Mar 2019

The author would like to thank Anonymous referee #1for the constructive and helpful suggestions on this manuscript.
We replied to 9 main comments and 9 minor comments

**Main Comment**

**1-3**. this paper is about the testing of the GEMS retrieval algorithm on OMI radiances, and the subsequent comparison with ozonesondes. This is interesting and worthwhile study in the run-up to the launch of GEMS. I see the paper as having two main purposes with respect to GEMS. First, it is an exercise of the retrieval algorithm on real OMI data. The fact that this is successful gives confidence that the retrieval is ready to receive the first GEMS data after launch. However, quantitative verification of the retrieval performance is harder, and the discussion of the GEMS retrieval algorithm performance on OMI radiances against sondes, compared with the OMI-algorithm retrievals against the same sondes should be expanded. The second purpose is to identify those ozonesonde measurements that might be good for GEMS validation, in as much as the work here suggests that they are useful or not for OMI validation. If there is a cross-verification here, it is really about OMI validation between OMI and the radiosondes. The fact that the GEMS algorithm is used to process the OMI radiances does not change this, especially with comparisons that should adequately account for how a priori profile and smoothing error assumptions differ between the GEMS and usual OMI algorithms. As such, the title of the manuscript does not clearly describe what is done in the paper, and I suggest that the authors modify the title to better reflect the above two goals.

 ▪ As indicated by this reviewer, the simulated GEMS retrievals are similar to OMI retrievals (PROFOZ), but with different and better implementations. Therefore the cross-verification performed in this paper is actually close to OMI validation than GEMS validation. The smoothing errors between OMI and GEMS have been addressed in Bak et al. (2013), indicating that GEMS will provide the comparable ozone profile information below ~ 22 km and the reduced information up to ~ 40 km. The tropopause-based ozone profile climatology is implemented as a priori ozone information for GEMS. However, the goal of this paper is not to detail the difference between OMI and GEMS ozone profile retrievals, but to evaluate the simulated GEMS tropospheric ozone retrievals using the limited UV information (300-330 nm) and to prepare the good reference dataset for GEMS validation. The quality comparison of simulated GEMS retrievals with OMI retrievals are additionally performed to demonstrate the confidence of the presented GEMS ozone profile algorithm. According to reviewer's suggestion, the title is changed to "Evaluation of GEMS tropospheric ozone retrieval performance using OMI data and validation ozonesonde dataset Over Northeast Asia" to clearly reflect what we did through this work.

**4**. **Section 2.1** describes the retrieval algorithm applied to the OMI radiances. A discussion should be included as to how the retrieval algorithm characteristics are expected to change for the GEMS radiances.

- GEMS is a scanning Uv/Visible spectrometer with a single UV-enhanced CCD for the spectral measurements of 300-500 nm (FWHM: 0.6 nm, spectral sampling: 0.2 nm) with at least comparable radiometric/wavelength accuracy (4% including light source uncertainty/0.01 nm) as OMI. However, GEMS data processing is expected to be different to OMI mainly in two ways: 1) OMI use a depolarizer to scramble the polarization of light. However, GEMS requires polarization factor < 2% and adopts polarization correction using RTM-based look-up table and pre-flight measured polarization characterization at the level 1b data processing. The GEMS polarization correction is less accurate and hence additional fitting process might be required in the level 2 data processing, especially for ozone profiles which the retrieval sensitivity to the polarization error is expected to be more significant compared to other trace-gases. 2) GEMS has a capability to perform diurnal observation and hence the diurnal meteorological input data are required to account for the temperature dependent Huggins band ozone absorption. Hence the numerical weather prediction (NWP) model analysis data will be transferred to the GEMS science data processing center (SDPC). This part is clarified in the Section 2.1 of the revised manuscript.

**5**. **Section 3.1**. The discussion of the differences between satellite/sonde agreements at the different sites is interesting. In addition to the differences between the sonde characteristics and reliabilities, one might expect greater standard deviations at sites that are polluted and/or show greater variability in ozone loadings due to meteorology. Some further discussion would be useful about the chemical-transport environment before eliminating sites from potential GEMS validation based on instrumentation/ experimental method arguments alone. It would also help this reader if the current dense text were broken up into descriptions of the various reasons for good/bad agreement.

- We have added the figure 4 in which the seasonal mean and standard deviations of ozonesonde measurements are presented to see the stability and characteristics of ozonesonde measurements at each site. Instabilities of measurements are apparently observed from New Delhi ozonesondes. High surface ozone concentration at Trivandrum in summer is believed to be caused by measurement errors because low level of the pollutants has been reported at this site under the geolocation and meteorological effects (Lal et al. 2000). Besides Trivandrum, Naha could be regarded as background sites according to low surface ozone and its precursor concentrations compared to neighboring stations (Fig. 2 and Fig. 4), and previous studies (Oltmans et al., 2004; Liu et al., 2002). In the lower troposphere high ozone concentrations are captured at Pohang, Tsukuba, and Sapporo in the summer due to enhanced photochemical production of ozone in daytime, whereas tropical sites, Naha, Hanoi, and Hong Kong show the ozone enhancement in spring mainly due to the biomass burning in Southeast Asia, with low ozone concentrations in summer due to the Asian monsoon and in winter due to the tropical air intrusion (Liu et al., 2002; Ogino et al., 2013). Singapore and Kuala lump are supposed to be severely polluted area, but ozone pollution is not clearly captured over the seasons. It could be explained by the observation time carried on in the morning. In addition, instabilities of Singapore measurements are noticeable such as abnormally large variability and very low ozone

concentration in the stratosphere. The effect of stratospheric intrusions on the ozone profile shape is dominant at the mid-latitudes (Pohang, Tsukuba, and Sapporo) during the spring and winter when the ozone pause goes down to 300 hPa, with the larger ozone variabilities in the lower stratosphere and upper troposphere, whereas ozonepause is placed around 100 hPa with much less variabilities of ozone over the pressure at other seasons. This discussion will be included in Section 3.

- Lal, S., Naja, M., and Subbaraya, B: Seasonal variations in surface ozone and its precursors over an urban site in India, Atmospheric Environment, Volume 34, Issue 17, 2000, Pages 2713-2724, 2000.
- Liu, H., D. J. Jacob, L. Y. Chan, S. J. Oltmans, I. Bey, R. M. Yantosca, J. M. Harris, B. N. Duncan, and R. V. Martin, Sources of tropospheric ozone along the Asian Pacific Rim: An analysis of ozonesonde observations, J. Geophys. Res., 107(D21), 4573, doi:10.1029/2001JD002005, 2002.
- Ogino, S.-Y., M. Fujiwara, M. Shiotani, F. Hasebe, J. Matsumoto, T. H. T. Hoang, and T. T. T. Nguyen (2013), Ozone variations over the northern subtropical region revealed by ozonesonde observations in Hanoi, J. Geophys. Res. Atmos., 118, 3245–3257, doi:10.1002/jgrd.50348.

[Figure]

**Fig**. 4. Seasonal mean (solid) and standard deviation (dashed) of ozonesonde soundings from 2005 to 2015 at 10 sites. 5 mPa is subtracted to standard deviations to fit in the given x-axis.

[Figure]

**Figure 2.** Geographic locations of the ozonesonde stations available since 2005 over the GEMS observation domain. The background map illustrates the OMI NO2 monthly mean in June 2015.

**6**. **Section 4.2 and Fig. 7**. The impact on correlation of smoothing or not smoothing the sonde profiles might be dependent on how close the GEMS retrieval a priori profile is to the sonde "truth". How do these compare and how do they vary between locations of good and poor comparison? How far does a priori profile go toward explaining the bias?

- To explain this impact, the comparison between GEMS a priori and ozonesondes is presented below, similarly to Fig.6 (profile comparison). This a priori information is taken from the tropopause-based ozone profile climatology (TB) which adjusts a monthly and zonal mean ozone profile with a daily tropopause height. Bak et al. (2013) demonstrated that TB based a priori better represents the ozone variabilities in the extra-tropical upper troposphere and lower stratosphere, especially during the winter and spring when the atmospheric status is strongly controlled by the dynamics. A priori information is very important to the quality of UV ozone profile retrievals, but the retrieved ozone profiles show much better agreement with ozonesondes than a priori ozone profiles, implying that the independent piece of information are available from these UV measurements. We can see that the biases seen in a priori are significantly reduced for most stations except for the Singapore station. Positive biases around 15 km in the a priori at the three mid-latitude sites still remain in the retrievals but with much smaller magnitude. Negative biases around ~17 km at other lower-latitude sites (except for Singapore) are almost eliminated in the retrievals.

[Figure]

S1. Same as Fig 6, but for comparison of ozonesondes and a TB-based priori ozone profiles.

**7-8**. Section 4.2 should be split into another sub section at Line 354 that starts the discussion of the evaluation of the GEMS algorithm against the OMI algorithm. This should be presented as one of the main results sections of this paper: a quantitative evaluation of the GEMS algorithm against other widely used algorithms based on the same OMI radiances. Given the previous discussion in the paper of the various limitations of some of the sondes, it might additionally be useful to directly compare the results of the different retrieval algorithms and explain differences in results on the basis of different features of the algorithms. This would help make the case that the GEMS algorithm is performing as expected.

- In this paper, the ozonesonde measurements available in the GEMS domain are characterized and validated to better evaluate the performance of the GEMS ozone profile algorithm. The accuracy and precision of simulated GEMS ozone profiles are established against the selected true reference. Additionally, the consistent evaluation is performed for the existing OMI ozone products to check the confidence of the GEMS retrieval algorithm, demonstrating that the comparable or better performance of GEMS ozone profile retrievals in the comparison with ozonesonde measurements. The different validation results between retrieval algorithms were discussed such as "GEMS algorithm is developed based on the heritages of the SAO ozone profile algorithm with several modifications. There are two main modifications: a priori ozone climatology was replaced with a tropopause-based ozone profile climatology to better represent the ozone variability in the tropopause. Irradiance spectra used to normalize radiance spectra and characterize instrument line shapes are prepared by taking 31-day moving average instead of climatological average to take into account for time-dependent instrument degradations. These modifications reduce somewhat spreads in deviations of satellite retrievals from sondes, especially in TCO comparison. KNMI retrievals systematically overestimate the tropospheric ozone by ~ 6 DU (Fig. 9.c), which corresponds to the positive biases of 2-4 % in the integrated total columns of KNMI profiles relative to Brewer observations (Bak et al., 2015). As mentioned in Bak et al. (2015), the systematic biases in ozone retrievals are less visible in SAO-

based retrievals (GEMS simulation, OMPROFOZ) as systematic components of measured spectra are taken into account for using an empirical correction called "soft calibration". The GEMS algorithm is very similar to the SAO algorithm except for the use of TB climatology and the impact of TB on the retrievals was discuss in detail in Bak et al. (2013). Also the comparison of SAO and KNMI algorithms were discussed in detail in Bak et al. (2015). So we think that it is more efficient to place the discussion related to Figures 7-9 in the same section and the direct comparison of GEMS and other OMI product is beyond the scope of this paper.

***9.*** The paper requires careful, and extensive, editing for English usage, and cut-paste typos, e.g. line 75, that should have been corrected before manuscript submission.

- This manuscript is going to be carefully revised though native English co-author before the submission of the revised manuscript.

**Minor Comments**

***1.*** Several times "GEMS" measurements are described. The word "simulated" should be added each time to avoid confusion

- In this revised manuscript, "GEMS measurements" was edited to " simulated GEMS measurements"

***2. Line 83:*** consistent perhaps, but not homogeneous as the authors point out in the text above.

- The indicated sentence was revised from "a homogenous, consistent ozonesonde" to "a consistent ozonesonde".

***4. Line 105:*** Instrument errors certainly, but also instrument design sensitivity.

- The indicated word, "instrumental errors″ was revised to "Instrument errors certainly, but also instrument design sensitivity″

***5. Line 106:*** Common geophysical conditions can reduce sensitivity, not just extreme

- The associated sentence was edited from "The impact of a priori information on retrievals become important  under certain geophysical conditions.

***6. Line 123***: Information may be limited but is a goal of the GEMS mission. This should be clarified.

- The GEMS mission was originally planned to develop the spectrometer for measuring the tropospheric pollutants, the spectral coverage of 300-500 nm satisfies to observe the tropospheric ozone as well as the lower/middle stratospheric ozone.

***7. Line 157***: Any more recent references to new measurement technique and instrumentation?

- More references (Thompson et al., 2017; Witte et al., 2017; 2018) are added in this sentence.

***8. Line 200***: How do these coincidence criteria for OMI and the sondes affect the results? What is the expected variation within the time and space windows? What is the representativeness uncertainty? How do these results here with the OMI comparison inform on the expected GEMS comparisons with hourly measurements at ∼7km resolution?

- As mentioned in Section 2.3, the coincidence criteria between satellite and ozonesonde are: ±1.0° in both longitude and latitude and ±12 hours in time and then the closest pixel is selected. The actual spatiotemporal difference is much smaller than this criteria (57.5 km to 66.6 km, ~3 hours). The close collocation can significantly minimize the

effects of spatiotemporal variability on the comparison and therefore GEMS validation accuracy could be enhanced compared to OMI, which is newly included in the summary of this paper.

***9. Line 231***: Is "troposphere" written where is should be stratosphere?

- The indicated sentence was edited to "much coarser vertical resolution of 10-14 km in the troposphere and 7-11 km in the stratosphere".

---

## Referee Comment (RC2) · Anonymous Referee #2 · 24 May 2019

Main comments

This paper intends the validation of GEMS tropospheric ozone retrievals with respect to ozonesonde measurements, before the launch of GEMS. The following main issues need to be addressed:

1. GEMS ozone profile algorithm is applied to OMI BUV measurements. It should be explained why GEMS radiances has not been simulated instead and what is the impact of using LEO measurements for a GEO instrument.

2. The use of OMI measurements makes the title of the paper confusing as the validation is of OMI using GEMS algorithm, but not of GEMS. This needs to be changed.

3. Simulated GEMS retrievals are used to verify the ozonesonde observations, ie., to identify the good stations, and in turn, these stations are use to validate the simulated GEMS retrievals. Using this approach it is hard to expect bad results for the simulated GEMS retrievals. The ozonesonde observations should be considered as the truth, and if they need to be validated and screened, this should be done using an independent dataset, but not the same dataset that we intend to validate, in this case the simulated GEMS retrievals.

4. According to the results shown, the time frame established of +-12 hours seems too large for the evaluation of tropospheric ozone, especially for mid-latitudes location where a stronger daily cycle can be found.

Minor comments

- Line 50: Satellite name should be Sentinel-4

- Line 75: "... have yet to be not been ..." please correct this

- Line 178: Among ECC stations

- Line 183: "Kula lump", please correct. Also all along the paper, the name of this station is written in different ways (Kuala lump, Kuala Lumpur). Please homogenize the station names in the text, figures and tables.

- Line 221: biased -> bias

- Line 225: Please specify the units

- Line 231: troposphere -> stratosphere

- Line 234: Should be photons?

- Line 242: xa should be placed after (1-A)

- Line 282: Please rephrase, maybe "of" -> "with values ranging from"

- Line 290: "Japanese stations" or "stations from Japan". Same in Line 296

- Line 314: Please unify or explain the differences between LT, LS and LST across the paper

- Line 322: "oznesonde" -> "ozonesonde"

- Line 324: Please list stations after "mid-latitude" and refer to Figure justifying this and the following statements.

- Line 326: "- a few %" please rephrase this

- Line 338: 4.2 -> 3.2

- Line 358: "... gives the good information ..." please rephrase. SOC has not been defined

- Line 367: "espeically" -> "especially". "TCO" -> TOC

- Line 308: Shouldn't it be "latitudinally" as it is used in other parts of the manuscript? Same in Line 398 and Line 400 (in this case, why capital L?)

- Line 399: Extra s "is similarly"

- Figure 2: Latitudes and Longitudes are not correct

- Figure 3: Please explain what is CF(O) and CF(X). Even if no CF is applied to MF sondes, it would be interesting to add them in Figure 3.

- Figure 4: Please specify how you differentiate the different type of sondes. Is it using diamonds, full dots and empty dots? Which one is which? Also indicate what is the horizontal axes, eg. "time (years)"

- Figure 6: I would suggest rewritting the last sentence as follows "The relative difference (in %) is defined as 100 X (SONDE AK – GEMS) / (A priori)". Why is multiplied by 2?

- Figure 7 and 8: Please replace TCO -> TOC and SCO -> SOC to be consistent with the text.

---

## Author Comment (AC2) · 9 Jun 2019

**Response to referee #2's comments**

The author would like to thank Anonymous referee #2 for the constructive and helpful suggestions on this manuscript.

We replied to 4 main comments and 25 minor comments

**Main Comment**

**C1**. GEMS ozone profile algorithm is applied to OMI BUV measurements. It should be explained why GEMS radiances has not been simulated instead and what is the impact of using LEO measurements for a GEO instrument.

**R1**. The development of the GEMS L2 algorithm has been in progress with OMI measurements because the simulation of the GEMS radiances using the forward model has not been fully implemented. Two main differences in GEMS and LEO (OMI) data processing could be expected: 1) OMI use a depolarizer to scramble the polarization of light. However, GEMS has polarization sensitivity (required to be less than < 2%) and performs polarization correction using RTM-based look-up table of atmospheric polarization state and pre-flight characterization of polarization sensitivity in the level 0 to 1b data processing. The GEMS polarization correction is less accurate and hence additional fitting process might be required in the level 2 data processing, especially for ozone profiles that have more significant retrieval sensitivity to the polarization error compared to other trace-gases. 2) GEMS has a capability to perform diurnal observation and hence the diurnal meteorological input data are required to account for the temperature dependent Huggins band ozone absorption. Hence the numerical weather prediction (NWP) model analysis data will be transferred to the GEMS science data processing center (SDPC). This response has been also included in the revised manuscript, also according to the comment #4 from reviewer 1.

**C2**. The use of OMI measurements makes the title of the paper confusing as the validation is of OMI using GEMS algorithm, but not of GEMS. This needs to be changed.

**R2**. This reply is also corresponding to comment #1 from reviewer 1, the title of this paper is changed to "Cross-Evaluation of GEMS tropospheric ozone retrieval performance using OMI data and the use of ozonesonde dataset over East Asia for validation".

**C3**. Simulated GEMS retrievals are used to verify the ozonesonde observations, i.e., to identify the good stations, and in turn, these stations are used to validate the simulated GEMS retrievals. Using this approach it is hard to expect bad results for the simulated GEMS retrievals. The ozonesonde observations should be considered as the truth, and if they need to be validated and screened, this should be done using an independent dataset, but not the same dataset that we intend to validate, in this case the simulated GEMS retrievals.

**R3**. Understanding the quality of the reference dataset and then selecting a good reference is a very important process in validating satellite or other in-situ measurements and then in better characterizing the retrieval accuracy and error. Satellite measurements of tropospheric ozone have previously been utilized to disclose problems in ozonesonde observations (e.g., Liu et al., 2006; Huang et al., 2018). We are also using retrievals here to identify ozonesonde measurements with significant errors. However, the station-to-station based quality control has not been typically applied in previous validation works. The figure below demonstrates how much the accuracy of the simulated GEMS retrievals from OMI measurements is underestimated if the station-to-station based quality control is not applied. We also apply the parallel validation for two independent OMI ozone profile products, OMPROFOZ and OMO3PR, respectively, demonstrating that our ozone retrievals are in comparable or better agreement with ozonesondes. As we mentioned in R1 to C1, the simulation of the GEMS radiances using the forward model has not been fully implemented.

[Figure]

**S1. Same as Figure 7, but for including all ECC measurements.**

**C4**. According to the results shown, the time frame established of +-12 hours seems too large for the evaluation of tropospheric ozone, especially for mid-latitudes location where a stronger daily cycle can be found.

**R4**. Based on the previous papers, the collocation of satellite pixel to ozonesonde stations have been performed within 6 to 24 hours. As clarified in Sect. 2.3 such as "The coincidence criteria between satellite and ozonesonde are: $\pm 1.0^{\circ}$ in both longitude and latitude and $\pm 12$ hours in time and then the closest pixel is selected. The Aura satellite carrying OMI crosses the equator always at ~ 1:45 pm LT and thereby OMI measurements are closely collocated within 3 hours to ozonesonde soundings measured in afternoon (1-3 pm LS)," OMI measurements are closely collocated within 3 hours to ozonesonde soundings measured in afternoon (1-3 pm) from Japanese stations, Pohang, Hong Kong, Hanoi, and Trivandrum. In this paper, the time collocation criterion is set to be 12 hours to include other stations existing over the GEMS domain.

**Minor comments**

**C1**. Line 50: Satellite name should be Sentinel-4

**R1.** This name has been corrected to "Sentinel-4"

**C2**. Line 75: ":: : have yet to be not been ..." please correct this

**R2**. It has been corrected to "has not been".

**C3**. Line 178: Among ECC stations

**R3**. It has been corrected to "Among ECC stations".

**C4**. Line 183: "Kula lump", please correct. Also all along the paper, the name of this station is written in different ways (Kuala lump, Kuala Lumpur). Please homogenize the station names in the text, figures and tables.

**R4**. We carefully checked what this reviewer indicated. This station name has been corrected to "Kuala Lumpur" across the manuscript.

**C5**. Line 221: biased -> bias

**R5**. It has been corrected to "bias".

**C6**. Line 225: Please specify the units

**R6**. RMS does not have the unit and thereby "RMS (i.e., root mean square of fitting residuals relative to measurement errors) less than 3" has been kept in the revised manuscript.

**C7**. Line 231: troposphere -> stratosphere

**R7**. It has been corrected to "stratosphere".

**C8**. Line 234: Should be photons?

**R8**. It has been revised to "photons".

**C9**. Line 242: xa should be placed after (1-A)

**R9**. Eq. 3 has been revised to "$\hat{x}_{sonde} = A \cdot x_{sonde} + (1 - A)x_a$"

**C10**. Line 282: Please rephrase, maybe "of" -> "with values ranging from"

**R10**. According to this comment, "satellite retrievals show the distinct seasonal TOC variations with the amplitude of ~ 35-40 DU" has been edited to "~ seasonal TOC variations with the values ranging from ~35 to ~ 40 DU"

**C11.** Line 290: "Japanese stations" or "stations from Japan". Same in Line 296.

**R11.** It has been revised to "stations from Japan"

**C12**. Line 314: Please unify or explain the differences between LT, LS and LST across the paper

**R12**. There is no difference. It has been unified to "LT (Local time)"

**C13**. Line 322: "oznesonde" -> "ozonesonde"

**R13**. This word has been corrected.

**C14**. Line 324: Please list stations after "mid-latitude" and refer to Figure justifying this and the following statements.

**R14**. It has been clarified such as "mid-latitude (Pohang, Tsukuba, and Sapporo)"

**C15**. Line 326: "- a few %" please rephrase this

**R15**. It has been corrected to "a few percent"

**C16**. Line 338: 4.2 -> 3.2.

**R16**. It has been changed to "3.2".

**C17**. Line 358: "... gives the good information ..." please rephrase. SOC has not been defined

**R17**. It has been corrected to "gives the good information on Stratospheric Ozone Column (SOC)"
"

**C18**. Line 367: "espeically" -> "especially". "TCO" -> TOC.

**R18**. The relevant sentence has been corrected to "especially in the TOC comparison"

**C19**. Line 308: Shouldn't it be "latitudinally" as it is used in other parts of the manuscript? Same in Line 398 and Line 400 (in this case, why capital L?)

**R19**. "latitudinally" was used at lines, 27, 286, 308, 398, and 400, respectively. These have been revised as followings,

- At 27, "compared to latitudinally adjacent stations with Carbon Iodine (CI) and Electrochemical Condensation Cell (ECC)." to "Carbon Iodine (CI) and Electrochemical Condensation Cell (ECC) dataset measured in similar latitude regime"
- At 308, "latitudinally adjacent station, Hong Kong" to "neighboring station, Hong Kong"
- At 398, "latitudinally adjacent Japanese 398 ECC measurements at Tsukuba and Sapporo" to "Japanese ECC measurements at Tsukuba and Sapporo located in mid-latitudes (> 30 °)"
- At 400, at Naha and Hong Kong stations located in similar latitude regime.

**C20**. Line 399: Extra s "is similarly"

**R20.** This indicated one (is s similarly) has been corrected (is similarly)

**C21**. Figure 2: Latitudes and Longitudes are not correct.

**R21**. This figure has been revised.

**C22**. Figure 3: Please explain what is CF(O) and CF(X). Even if no CF is applied to MF sondes, it would be interesting to add them in Figure 3.

**R22**. To clarify, the legend in the figure has been revised to "Solid: with CF, Dash: w/o CF". The corresponding caption has been revised to "Effect of applying a correction factor (CF) to (a) ECC and (b) CI ozonesonde measurements, respectively on comparisons with simulated GEMS ozone profile retrievals. Solid and Dashed lines represent the comparisons with and without applying a CF, respectively, at each Japanese station."

**C23**. Figure 4: Please specify how you differentiate the different type of sondes. Is it using diamonds, full dots and empty dots? Which one is which? Also indicate what is the horizontal axes, eg. "time (years)"

**R23**. This figure has been revised to clarify the symbols and the title of x-axis.

**C24**. Figure 6: I would suggest rewritting the last sentence as follows "The relative difference (in %) is defined as 100 X (SONDE AK – GEMS) / (A priori)". Why is multiplied by 2?

**R24**. This equation has been corrected to "100 X (SONDE AK – GEMS) / (A priori)"

**C25**. Figure 7 and 8: Please replace TCO -> TOC and SCO -> SOC to be consistent with the text.

**R25**. This figure has been revised to accept this comment.

---

## Editor Decision (ED1)

**Cross-evaluation of GEMS tropospheric ozone retrieval performance using OMI data and the use of ozonesonde dataset over East Asia for validation**

Juseon Bak, Kang-Hyeon Baek, Jae-Hwan Kim, Xiong Liu, Jhoon Kim, and Kelly Chance

Dear authors,

After the two reviews and after reading the your answers and revised paper, I am glad to accept the revised paper for publication for AMT. However, before the final edition I would like you to consider some comments from me.

In the abstract: Please, to be consistent with the new title, change « cross-verification » by « cross evaluation ».

In sect. 2.3: please add the information of spectral resolution of GEMS and OMI to show the reader the differences between the two instruments.

Line 486: The end of the sentence is unclear to me. I would say the opposite that the ozonesonde has to verify the retrievals? Please consider also to cut it into two sentences.

Line 567-Line 745: space-born-> space borne and  « balloon-born » -> balloon-borne ? Check through the paper if necessary.

Line 675 : This is not clear to me what the vertical resolution represents here (10-14 km in the troposphere) and (7-11 km) in the stratosphere. Is it representing the width of the broad peak of the averaging kernel? (at different levels or for the columns).  If yes, I would suggest you to add a typical figure of averaging kernel of GEMS ozone and why not indicating the degree of freedom?

Line 592: There is a space missing between SST and 1.0 and please define SST and KI

Table 1: upper script d is missing in the caption.

All the best,

Jean-Luc Attié

---

## Author Response (AR2)

**a point-by-point response to the reviews**

We replied to 9 main comments and 9 minor comments hereafter.

**Main Comment**

**1-3**. this paper is about the testing of the GEMS retrieval algorithm on OMI radiances, and the subsequent comparison with ozonesondes. This is interesting and worthwhile study in the run-up to the launch of GEMS. I see the paper as having two main purposes with respect to GEMS. First, it is an exercise of the retrieval algorithm on real OMI data. The fact that this is successful gives confidence that the retrieval is ready to receive the first GEMS data after launch. However, quantitative verification of the retrieval performance is harder, and the discussion of the GEMS retrieval algorithm performance on OMI radiances against sondes, compared with the OMI-algorithm retrievals against the same sondes should be expanded. The second purpose is to identify those ozonesonde measurements that might be good for GEMS validation, in as much as the work here suggests that they are useful or not for OMI validation. If there is a cross-verification here, it is really about OMI validation between OMI and the radiosondes. The fact that the GEMS algorithm is used to process the OMI radiances does not change this, especially with comparisons that should adequately account for how a priori profile and smoothing error assumptions differ between the GEMS and usual OMI algorithms. As such, the title of the manuscript does not clearly describe what is done in the paper, and I suggest that the authors modify the title to better reflect the above two goals.

- As indicated by this reviewer, the simulated GEMS retrievals are similar to OMI retrievals (PROFOZ), but with different and better implementations. Therefore the cross-verification performed in this paper is actually close to OMI validation than GEMS validation. The smoothing errors between OMI and GEMS have been addressed in Bak et al. (2013), indicating that GEMS will provide the comparable ozone profile information below ~ 22 km and the reduced information up to ~ 40 km. The tropopause-based ozone profile climatology is implemented as a priori ozone information for GEMS. However, the goal of this paper is not to detail the difference between OMI and GEMS ozone profile retrievals, but to evaluate the simulated GEMS tropospheric ozone retrievals using the limited UV information (300-330 nm) and to prepare the good reference dataset for GEMS validation. The quality comparison of simulated GEMS retrievals with OMI retrievals are additionally performed to demonstrate the confidence of the presented GEMS ozone profile algorithm. According to reviewer's suggestion, the title is changed to "Cross-evaluation of GEMS tropospheric ozone retrieval performance using OMI data and the use of ozonesonde dataset over East Asia for validation" to clearly reflect what we did through this work.

**4**. **Section 2.1** describes the retrieval algorithm applied to the OMI radiances. A discussion should be included as to how the retrieval algorithm characteristics are expected to change for the GEMS radiances.

- GEMS is a scanning Uv/Visible spectrometer with a single UV-enhanced CCD for the spectral measurements of 300-500 nm (FWHM: 0.6 nm, spectral sampling: 0.2 nm) with at least comparable radiometric/wavelength accuracy (4% including light source uncertainty/0.01 nm) as OMI. However, GEMS data processing is expected to be different tofrom OMI mainly in two ways: 1) OMI uses a depolarizer to scramble the polarization of light. However, GEMS has polarization sensitivity (required to be less than 2%) and performs polarization correction using an RTM-based look-up table of atmospheric polarization state and pre-flight characterization of polarization instrument polarization sensitivity in the level 0 to 1b data processing. The GEMS polarization correction is less accurate and hence additional fitting process might be required in the level 2 data processing, especially for ozone profiles that have more significant retrieval sensitivityare more sensitive to the polarization error compared to other trace-gases. 2) GEMS has a capability to perform diurnal observations and hence the diurnal meteorological input data are required to account for the temperature dependent Huggins band ozone absorption. Hence, the numerical weather prediction (NWP) model analysis data will be transferred to the GEMS Sscience Ddata Pprocessing Ccenter (SDPC). This part is clarified in the Section 2.1 of the revised manuscript.

**5**. **Section 3.1**. The discussion of the differences between satellite/sonde agreements at the different sites is interesting. In addition to the differences between the sonde characteristics and reliabilities, one might expect greater standard deviations at sites that are polluted and/or show greater variability in ozone loadings due to meteorology. Some further discussion would be useful about the chemical-transport environment before eliminating sites from potential GEMS validation based on instrumentation/ experimental method arguments alone. It would also help this reader if the current dense text were broken up into descriptions of the various reasons for good/bad agreement.

- We have added the figure 4 in which the seasonal mean and standard deviations of ozonesonde measurements are presented to see the stability and characteristics of ozonesonde measurements at each site. Instabilities of measurements are observed from New Delhi ozonesondes. High surface ozone concentrations at Trivandrum in summer are believed to be caused by measurement errors because low levels of pollutants have been reported at this site under the geolocation and meteorological effects (Lal et al. 2000). Besides Trivandrum, Naha could be regarded as background sites according to low surface ozone and its precursor concentrations compared to neighboring stations (Fig. 2 and Fig. 4), and previous studies (Oltmans et al., 2004; Liu et al., 2002). In the lower troposphere high ozone concentrations are captured at

Pohang, Tsukuba, and Sapporo in the summer due to enhanced photochemical production of ozone in daytime, whereas tropical sites, Naha, Hanoi, and Hong Kong show the ozone enhancement in spring mainly due to biomass burning in Southeast Asia, with low ozone concentrations in summer due to the Asian monsoon and in winter due to the tropical air intrusion (Liu et al., 2002; Ogino et al., 2013). Singapore and Kuala lump are supposed to be severely polluted area, but ozone pollution is not clearly captured over the seasons. It could be explained by the morning observation time at these two stations. In addition, instabilities of Singapore measurements are noticeable, including abnormally large variability and very low ozone concentration in the stratosphere. The effect of stratospheric intrusions on the ozone profile shape is dominant at mid-latitudes (Pohang, Tsukuba, and Sapporo) during the spring and winter when the ozone pause goes down to 300 hPa, with larger ozone variabilities in the lower stratosphere and upper troposphere, whereas ozonepause is around 100 hPa with much less variability of ozone in other seasons. This discussion has been included in Section 3.

- Lal, S., Naja, M., and Subbaraya, B: Seasonal variations in surface ozone and its precursors over an urban site in India, Atmospheric Environment, Volume 34, Issue 17, 2000, Pages 2713-2724, 2000.
- Liu, H., D. J. Jacob, L. Y. Chan, S. J. Oltmans, I. Bey, R. M. Yantosca, J. M. Harris, B. N. Duncan, and R. V. Martin, Sources of tropospheric ozone along the Asian Pacific Rim: An analysis of ozonesonde observations, J. Geophys. Res., 107(D21), 4573, doi:10.1029/2001JD002005, 2002.
- Ogino, S.-Y., M. Fujiwara, M. Shiotani, F. Hasebe, J. Matsumoto, T. H. T. Hoang, and T. T. T. Nguyen (2013), Ozone variations over the northern subtropical region revealed by ozonesonde observations in Hanoi, J. Geophys. Res. Atmos., 118, 3245–3257, doi:10.1002/jgrd.50348.

[Figure]

**Fig**. 4. Seasonal mean (solid) and standard deviation (dashed) of ozonesonde soundings from 2005 to
2015 at 10 sites. 5 mPa is subtracted to standard deviations to fit in the given x-axis.

[Figure]

**Figure 2.** Geographic locations of the ozonesonde stations available since 2005 over the GEMS observation domain. The background map illustrates the OMI $NO_2$ monthly mean in June 2015.

**6**. **Section 4.2 and Fig. 7**. The impact on correlation of smoothing or not smoothing the sonde profiles might be dependent on how close the GEMS retrieval a priori profile is to the sonde "truth". How do these compare and how do they vary between locations of good and poor comparison? How far does a priori profile go toward explaining the bias?

- To explain this impact, the comparison between GEMS a priori and ozonesondes is presented below, similarly to Fig.6 (in old manuscript, it is Fig. 7 in the revised one). This a priori information is taken from the tropopause-based ozone profile climatology (TB) which adjusts a monthly and zonal mean ozone profile with a daily tropopause height. Bak et al. (2013) demonstrated that TB based a priori better represents the ozone variabilities in the extra-tropical upper troposphere and lower stratosphere, especially during the winter and spring when the atmospheric status is strongly controlled by the dynamics. A priori information is very important to the quality of UV ozone profile retrievals, but the retrieved ozone profiles show much better agreement with ozonesondes than a priori ozone profiles, implying that the independent piece of information are available from these UV measurements. We can see that the biases seen in a priori are significantly reduced for most stations except for the Singapore station. Positive biases around 15 km in the a priori at the three mid-latitude sites still remain in the retrievals but with much smaller magnitude. Negative biases around ~17 km at other lower-latitude sites (except for Singapore) are almost eliminated in the retrievals.

[Figure]

S1. Same as Fig 6, but for comparison of ozonesondes and a TB-based priori ozone profiles.

**7-8**. Section 4.2 should be split into another sub section at Line 354 that starts the discussion
of the evaluation of the GEMS algorithm against the OMI algorithm. This should be presented
as one of the main results sections of this paper: a quantitative evaluation of the GEMS
algorithm against other widely used algorithms based on the same OMI radiances. Given the
previous discussion in the paper of the various limitations of some of the sondes, it might
additionally be useful to directly compare the results of the different retrieval algorithms and
explain differences in results on the basis of different features of the algorithms. This would
help make the case that the GEMS algorithm is performing as expected.

▪    In this paper, the ozonesonde measurements available in the GEMS domain are
   characterized and validated to better evaluate the performance of the GEMS ozone
   profile algorithm. The accuracy and precision of simulated GEMS ozone profiles are
   established against the selected true reference. Additionally, the consistent evaluation
   is performed for the existing OMI ozone products to check the confidence of the
   GEMS retrieval algorithm, demonstrating that the comparable or better performance
   of GEMS ozone profile retrievals in the comparison with ozonesonde measurements.
   The different validation results between retrieval algorithms were discussed such as
   "GEMS algorithm is developed based on the heritages of the SAO ozone profile
   algorithm with several modifications. There are two main modifications: a priori ozone
   climatology was replaced with a tropopause-based ozone profile climatology to better
   represent the ozone variability in the tropopause. Irradiance spectra used to normalize
   radiance spectra and characterize instrument line shapes are prepared by taking 31-day
   moving average instead of climatological average to take into account for time-
   dependent instrument degradations. These modifications reduce somewhat spreads in
   deviations of satellite retrievals from sondes, especially in TCO comparison. KNMI
   retrievals systematically overestimate the tropospheric ozone by ~ 6 DU (Fig. 9.c), which corresponds to the positive biases of 2-4 % in the integrated total columns of
KNMI profiles relative to Brewer observations (Bak et al., 2015). As mentioned in Bak
et al. (2015), the systematic biases in ozone retrievals are less visible in SAO-based
retrievals (GEMS simulation, OMPROFOZ) as systematic components of measured
spectra are taken into account for using an empirical correction called "soft
calibration". The GEMS algorithm is very similar to the SAO algorithm except for the
use of TB climatology and the impact of TB on the retrievals was discuss in detail in
Bak et al. (2013). Also the comparison of SAO and KNMI algorithms were discussed
in detail in Bak et al. (2015). So we think that it is more efficient to place the discussion
related to Figures 7-9 in the same section and the direct comparison of GEMS and
other OMI product is beyond the scope of this paper.
▪

**9.** The paper requires careful, and extensive, editing for English usage, and cut-paste typos, e.g.
line 75, that should have been corrected before manuscript submission.
▪ This manuscript is going to be carefully revised though native English co-author
before the submission of the revised manuscript.

**Minor Comments**

**1.** Several times "GEMS" measurements are described. The word "simulated" should
be added each time to avoid confusion
▪ In this revised manuscript, "GEMS measurements" was edited to " simulated GEMS
measurements"
**2. Line 83:** consistent perhaps, but not homogeneous as the authors point out in the text above.
▪ The indicated sentence was revised from "a homogenous, consistent ozonesonde" to
"a consistent ozonesonde".
**4. Line 105:** Instrument errors certainly, but also instrument design sensitivity.

▪ The indicated word, "instrumental errors" was revised to "Instrument errors certainly, but also instrument design sensitivity"
**5. Line 106:** Common geophysical conditions can reduce sensitivity, not just extreme
▪ The associated sentence was edited from "The impact of a priori information on
retrievals become important  under
certain geophysical conditions.
**6. Line 123**: Information may be limited but is a goal of the GEMS mission. This should be
clarified.
▪ The GEMS mission was originally planned to develop the spectrometer for measuring
the tropospheric pollutants, the spectral coverage of 300-500 nm satisfies to observe
the tropospheric ozone as well as the lower/middle stratospheric ozone.
**7. Line 157**: Any more recent references to new measurement technique and instrumentation?
▪ More references (Thompson et al., 2017; Witte et al., 2017; 2018) are added in this
sentence.

**8. Line 200**: How do these coincidence criteria for OMI and the sondes affect the results? What
is the expected variation within the time and space windows? What is the representativeness
uncertainty? How do these results here with the OMI comparison inform on the expected
GEMS comparisons with hourly measurements at ~7km resolution?

- As mentioned in Section 2.3, the coincidence criteria between satellite and ozonesonde
  are: ±1.0º in both longitude and latitude and ±12 hours in time and then the closest
  pixel is selected. The actual spatiotemporal difference is much smaller than this criteria
  (57.5 km to 66.6 km, ~3 hours). The close collocation can significantly minimize the
  effects of spatiotemporal variability on the comparison and therefore GEMS validation
  accuracy could be enhanced compared to OMI, which is newly included in the
  summary of this paper (The impact of spatiotemporal variability on the comparison
  will be much reduced for GEMS due to its higher spatiotemporal resolution (7 km x 8
  km @ Seoul, hourly) against OMI (48 km x 13 km @ nadir in UV1, daily).

**9. Line 231**: Is "troposphere" written where is should be stratosphere?

- The indicated sentence was edited to "much coarser vertical resolution of 10-14 km in
  the troposphere and 7-11 km in the stratosphere".

<hr>

**Response to referee #2's comments**

<hr>

We replied to 4 main comments and 25 minor comments

**Main Comment**

**C1**. GEMS ozone profile algorithm is applied to OMI BUV measurements. It should be explained why GEMS radiances has not been simulated instead and what is the impact of using

LEO measurements for a GEO instrument.

**R1**. The development of the GEMS L2 algorithm has been in progress with OMI measurements because the simulation of the GEMS radiances using the forward model has not been fully implemented. Two main differences in GEMS and LEO (OMI) data processing could be expected: 1) OMI use a depolarizer to scramble the polarization of light. However, GEMS has polarization sensitivity (required to be less than < 2%) and performs polarization correction using RTM-based look-up table of atmospheric polarization state and pre-flight characterization of polarization sensitivity in the level 0 to 1b data processing. The GEMS

polarization correction is less accurate and hence additional fitting process might be required in the level 2 data processing, especially for ozone profiles that have more significant retrieval sensitivity to the polarization error compared to other trace-gases. 2) GEMS has a capability to perform diurnal observation and hence the diurnal meteorological input data are required to account for the temperature dependent Huggins band ozone absorption. Hence the numerical weather prediction (NWP) model analysis data will be transferred to the GEMS science data processing center (SDPC). This response has been also included in the revised manuscript, also according to the comment #4 from reviewer 1.

**C2**. The use of OMI measurements makes the title of the paper confusing as the validation is of OMI using GEMS algorithm, but not of GEMS. This needs to be changed.

**R2**. This reply is also corresponding to comment #1 from reviewer 1, the title of this paper is changed to "Cross-evaluation of GEMS tropospheric ozone retrieval performance using OMI data and the use of ozonesonde dataset over East Asia for validation".

**C3**. Simulated GEMS retrievals are used to verify the ozonesonde observations, i.e., to identify the good stations, and in turn, these stations are used to validate the simulated GEMS retrievals. Using this approach it is hard to expect bad results for the simulated GEMS retrievals. The ozonesonde observations should be considered as the truth, and if they need to be validated and screened, this should be done using an independent dataset, but not the same dataset that we intend to validate, in this case the simulated GEMS retrievals.

**R3**. Understanding the quality of the reference dataset and then selecting a good reference is a very important process in validating satellite or other in-situ measurements and then in better characterizing the retrieval accuracy and error. Satellite measurements of tropospheric ozone have previously been utilized to disclose problems in ozonesonde observations (e.g., Liu et al., 2006; Huang et al., 2018). We are also using retrievals here to identify ozonesonde measurements with significant errors. However, the station-to-station based quality control has not been typically applied in previous validation works. The figure below demonstrates how much the accuracy of the simulated GEMS retrievals from OMI measurements is underestimated if the station-to-station based quality control is not applied. We also apply the parallel validation for two independent OMI ozone profile products, OMPROFOZ and OMO3PR, respectively, demonstrating that our ozone retrievals are in comparable or better agreement with ozonesondes. As we mentioned in R1 to C1, the simulation of the GEMS

radiances using the forward model has not been fully implemented.

[Figure]

**S1. Same as Figure 8, but for including all ECC measurements.**

**C4**. According to the results shown, the time frame established of +-12 hours seems too large for the evaluation of tropospheric ozone, especially for mid-latitudes location where a stronger daily cycle can be found.

**R4**. Based on the previous papers, the collocation of satellite pixel to ozonesonde stations have been performed within 6 to 24 hours. As clarified in Sect. 2.3 such as "The coincidence criteria between satellite and ozonesonde are: ±1.0º in both longitude and latitude and ±12 hours in
time and then the closest pixel is selected. The Aura satellite carrying OMI crosses the equator
always at ~ 1:45 pm LT and thereby OMI measurements are closely collocated within 3 hours
to ozonesonde soundings measured in afternoon (1-3 pm LS),” OMI measurements are closely
collocated within 3 hours to ozonesonde soundings measured in afternoon (1-3 pm) from
Japanese stations, Pohang, Hong Kong, Hanoi, and Trivandrum. In this paper, the time
collocation criterion is set to be 12 hours to include other stations existing over the GEMS
domain.
**Minor comments**
**C1**. Line 50: Satellite name should be Sentinel-4
**R1.** This name has been corrected to "Sentinel-4"
**C2**. Line 75: ":: : have yet to be not been ..." please correct this
**R2**. It has been corrected to "has not been".
**C3**. Line 178: Among ECC stations
**R3**. It has been corrected to "Among ECC stations".
**C4**. Line 183: "Kula lump", please correct. Also all along the paper, the name of this station is
written in different ways (Kuala lump, Kuala Lumpur). Please homogenize the station names
in the text, figures and tables.
**R4**. We carefully checked what this reviewer indicated. This station name has been corrected
to "Kuala Lumpur" across the manuscript.
**C5**. Line 221: biased -> bias
**R5**. It has been corrected to "bias".
**C6**. Line 225: Please specify the units
**R6**. RMS does not have the unit and thereby "RMS (i.e., root mean square of fitting residuals
relative to measurement errors) less than 3" has been kept in the revised manuscript.

**C7**. Line 231: troposphere -> stratosphere

**R7**. It has been corrected to "stratosphere".

**C8**. Line 234: Should be photons?

**R8**. It has been revised to "photons".

**C9**. Line 242: xa should be placed after (1-A)

**R9**. Eq. 3 has been revised to "$\hat{x}_{sonde} = A \cdot x_{sonde} + (1 - A)x_a$"

**C10**. Line 282: Please rephrase, maybe "of" -> "with values ranging from"

**R10**. According to this comment, "satellite retrievals show the distinct seasonal TOC variations with the amplitude of ~ 35-40 DU" has been edited to "~ seasonal TOC variations with the values ranging from ~35 to ~ 40 DU"

**C11.** Line 290: "Japanese stations" or "stations from Japan". Same in Line 296.

**R11.** It has been revised to "stations from Japan"

**C12**. Line 314: Please unify or explain the differences between LT, LS and LST across the paper

**R12**. There is no difference. It has been unified to "LT (Local time)"

**C13**. Line 322: "oznesonde" -> "ozonesonde"

**R13**. This word has been corrected.

**C14**. Line 324: Please list stations after "mid-latitude" and refer to Figure justifying this and the following statements.

**R14**. It has been clarified such as "mid-latitude (Pohang, Tsukuba, and Sapporo)"

**C15**. Line 326: "- a few %" please rephrase this

**R15**. It has been corrected to "a few percent"

**C16**. Line 338: 4.2 -> 3.2.

**R16**. It has been changed to "3.2".

**C17**. Line 358: "... gives the good information ..." please rephrase. SOC has not been defined

**R17**. It has been corrected to "gives the good information on Stratospheric Ozone Column (SOC)"

"

**C18**. Line 367: "espeically" -> "especially". "TCO" -> TOC.

**R18**. The relevant sentence has been corrected to "especially in the TOC comparison"

**C19**. Line 308: Shouldn't it be "latitudinally" as it is used in other parts of the manuscript?

Same in Line 398 and Line 400 (in this case, why capital L?)

**R19**. "latitudinally" was used at lines, 27, 308, 398, and 400, respectively. These have been revised as followings,

-    At 27, "compared to latitudinally adjacent stations with Carbon Iodine (CI) and

Electrochemical Condensation Cell (ECC)." to "Carbon Iodine (CI) and Electrochemical

Condensation Cell (ECC) dataset measured in similar latitude regime"

-    At 308, "latitudinally adjacent station, Hong Kong" to "neighboring station, Hong Kong"

-    At 398, "latitudinally adjacent Japanese 398 ECC measurements at Tsukuba and Sapporo"

to "Japanese ECC measurements at Tsukuba and Sapporo located in mid-latitudes ($> 30°$)"

-    At 400, at Naha and Hong Kong stations located in similar latitude regime.

**C20**. Line 399: Extra s "is similarly"

**R20.** This indicated one (is s similarly) has been corrected (is similarly)

**C21**. Figure 2: Latitudes and Longitudes are not correct.

**R21**. This figure has been revised.

**C22**. Figure 3: Please explain what is CF(O) and CF(X). Even if no CF is applied to MF sondes, it would be interesting to add them in Figure 3.

**R22**. To clarify, the legend in the figure has been revised to "Solid: with CF, Dash: w/o CF".

The corresponding caption has been revised to "Effect of applying a correction factor (CF) to (a) ECC and (b) CI ozonesonde measurements, respectively on comparisons with simulated

GEMS ozone profile retrievals. Solid and Dashed lines represent the comparisons with and without applying a CF, respectively, at each Japanese station."

**C23**. Figure 4: Please specify how you differentiate the different type of sondes. Is it using diamonds, full dots and empty dots? Which one is which? Also indicate what is the horizontal axes, eg. "time (years)"

**R23**. This figure has been revised to clarify the symbols and the title of x-axis.

**C24**. Figure 6: I would suggest rewritting the last sentence as follows "The relative difference (in %) is defined as 100 X (SONDE AK – GEMS) / (A priori)". Why is multiplied by 2?

**R24**. This equation has been corrected to "100 X (SONDE AK – GEMS) / (A priori)"

**C25**. Figure 7 and 8: Please replace TCO -> TOC and SCO -> SOC to be consistent with the text.

**R25**. This figure has been revised to accept this comment.

**Response to editor #1's comments**

**C1.** In the abstract: Please, to be consistent with the new title, change « cross-verification »
by « cross Evaluation
**R2**. The word of "verification" has been changed to "evaluation" in the abstract.

**C2**. In sect. 2.3: please add the information of spectral resolution of GEMS and OMI to show
the reader the differences between the two instruments.
**R2**. To specify what this comment points out, "It is desgined to provide hyperspectral
radiances at a spectral resolution of 0.6 nm and spectral intervals of 0.2 nm, which are also
similar to OMI (spectral resolution: 0.42-0.63 nm, sampling rate: 0.14-0.33 nm/pixel)." has
been added in sect.2.1.

**C3**. Line 486: The end of the sentence is unclear to me. I would say the opposite that the
ozonesonde has to verify the retrievals? Please consider also to cut it into two sentences.
**R3**. Ultimately, ozonesonde dataset are used to qualtify the accuracy of simulated GEMS
retrievals, but as shown in this paper, ozonesonde measurements are not consistent each other
due to inconsistencies in the spatial and temporal irregularities in instrument types,
manufacturers, operating procedures, and correction strategies. Therefore we identify inconsistent ozonesonde data period/sites with respect to a better consistent GEMS simulated
retrievals because single data processing is applied to the entire satellite measurements
measured by single instrument. For clearfication, "that is, simulated GEMS retrievals using
OMI data retrievals ones areareis used to verify the ozonesonde observations." has been
deleted because the previous sentence is enough to specify what we point out.
**C4**. Line 567-Line 745: space-born-> space borne and « balloon-born » -> balloon-borne ?
Check through the paper if necessary.
**R4**. The mis-typed word, "born" has been corrected to "borne".
**C5**. Line 675 : This is not clear to me what the vertical resolution represents here (10-14 km in
the troposphere) and (7-11 km) in the stratosphere. Is it representing the width of the broad
peak of the averaging kernel? (at different levels or for the columns). If yes, I would suggest
you to add a typical figure of averaging kernel of GEMS ozone and why not indicating the
degree of freedom?
**R5.** The full width at half maximum of the main peak of each averaging kernel is an estimate
of the vertical resolutiona that altitude (Rodgers, 2000). Based on this definition, the vertical
resolultion of UV ozone profile retrievals is estimated as 10-14 km in the troposphere and (7-
11 km) in the stratosphere (e.g., Kroon et al., 2010; Liu et al., 2010). In associated paragraph,
I gave the general information on the vertical resolution of UV ozone profile retrievals to
emphasize on such a discrepancy with that of ozonesonde in-situ measurements and its impact
on the comparison between satellite retrievals and ozonesonde measurements. Therefore, I
added the references (Kroon et al., 2010; Liu et al., 2010) instead of adding a figure.
**C6.** Line 592: There is a space missing between SST and 1.0 and please define SST and KI.
**R6.** The space between SSI and version (e.g., 1.0) is removed consistenly in this manuscript.
To specify the acronyms of SSI and KI, the sentence where these acronyms are firstly used
been modified to "The the recommend recipes of the Standard Sensing Solution (SSI) are 1.0 %
potassium iodide (KI)/full buffer (SST1.0) and 2.0 % KI/no buffer (SST0.5) for the SPC and
EN-SCI sondes, respectively by the ASOPOS (Assessment for Standards on Operation
Procedures for Ozone Sondes) (Smit et al., 2012)"
**C7.** Table 1: upper script d is missing in the caption.
**R7.** It is included in this caption as "d Potassium Iodide (KI) cathode sensing solution type
(SST) implemented in ECC ozone sensors: SST 0.5 (0.5 % KI, half buffer), SST 1.0 (1.0 % KI,
full buffer), and SST 2.0 (2.0 % KI, no buffer). Singapore station changed it to SST 1.0 % as
of 2018"

**a list of all relevant changes**

1. All figures have been revised for better visibility, with newly included figure 4.

**a marked-up manuscript version**

[revised manuscript text omitted]